# MLLM-CL: Continual Learning for Multimodal Large Language Models

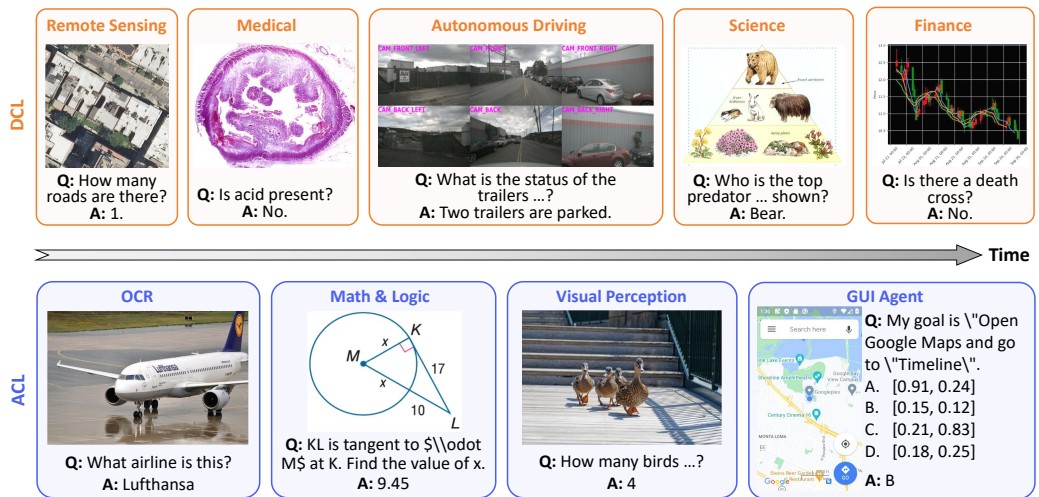

Figure 1: Demonstrations of MLLM-CL benchmark. It incorporates Domain Continual Learning (DCL), which adds domain-specific knowledge, and Ability Continual Learning (ACL), which improves fundamental abilities for multimodal large language models.

## ABSTRACT

Recent Multimodal Large Language Models (MLLMs) excel in vision-language understanding but face challenges in adapting to dynamic real-world scenarios that require continuous integration of new knowledge and skills. While continual learning (CL) offers a potential solution, existing benchmarks and methods suffer from critical limitations. In this paper, we introduce MLLM-CL, a novel benchmark encompassing domain and ability continual learning, where the former focuses on independently and identically distributed (IID) evaluation across evolving mainstream domains, whereas the latter evaluates on non-IID scenarios with new model abilities. Methodologically, we propose preventing catastrophic interference through parameter isolation and an MLLM-based routing mechanism. Extensive experiments demonstrate that our approach can integrate domain-specific knowledge and functional abilities with minimal forgetting, significantly outperforming existing methods. Our benchmark and code will be publicly available.

## 1 INTRODUCTION

Recent advancements in Multimodal Large Language Models (MLLMs) (Liu et al., 2024a; Chen et al., 2024b) have demonstrated remarkable capabilities in vision-language understanding. These models typically undergo supervised finetuning on carefully curated multi-task datasets, whereas real-world applications require continuous adaptation to evolving user requirements and dynamic data streams with shifting domain distributions. To incorporate new knowledge and skills, full retraining of large models is costly in both time and computing resources; besides, straightforward finetuning on novel tasks often results in catastrophic forgetting (McCloskey & Cohen, 1989; Zhai

et al., 2023). Therefore, for deployment in ever-changing environments, there is an urgent need to develop MLLMs capable of continually consolidating new skills while maintaining performance on prior tasks. Recently, a few studies (Chen et al., 2024a; Zeng et al., 2024; Cao et al., 2024; Guo et al., 2025a; He et al., 2023) have explored continual learning (CL) of MLLMs. However, current works still have key limitations in both benchmarks and methodologies, preventing them from effectively exploring CL in MLLMs.

Firstly, there is a lack of well-established benchmarks. Chen et al. (2024a) proposed the first continual instruction tuning benchmark for MLLMs comprising several downstream datasets, while some of them have already been learned during the early supervised finetuning (SFT) phase of MLLM. Huai et al. (2025) divided VQAv2 (Goyal et al., 2017) into several tasks and conducted continual instruction tuning directly from the LLaVA (Liu et al., 2023) base model. However, in real-world applications, continually learning subsets of a specific dataset is impractical, and it is unlikely to finetune an MLLM on downstream tasks without any SFT on general multimodal data. Moreover, those benchmarks only consider independently and identically distributed (IID) evaluation (the training and test sets are split from the same dataset), while the model would encounter non-IID inputs in practice.

Secondly, existing methods have notable limitations: (1) Some approaches share the same set of parameters for different tasks (Chen et al., 2024a; Huang et al., 2024). This might be suitable for a conventional class-incremental learning scenario where different tasks often belong to the same dataset. However, MLLMs often encounter inputs from various domains, and the inherent task conflicts (Wei et al., 2025; Yang et al., 2024) would lead to loss of plasticity during continual learning, particularly when handling heterogeneous modalities across divergent domains. (2) Parameter isolation methods have to determine which task-specific parameters to apply for a given input during inference. This selection is usually driven by simple hand-crafted similarity metrics (Zeng et al., 2024; Guo et al., 2025a), which can be unreliable when confronted with complex multimodal data, consequently undermining overall performance.

In this paper, we establish a novel benchmark MLLM-CL, which includes two practical settings, *i.e.*, domain continual learning (DCL) and ability continual learning (ACL), as shown in Fig. 1. Specifically, DCL aims to equip the model with domain-specific knowledge continually by learning and evaluating on several mainstream domains (remote sensing, medical, autonomous driving, science, and finance), where the training and test sets are IID. Differently, ACL focuses on incorporating fundamental abilities (OCR, math & logic, visual perception, and GUI agent), which are evaluated on non-IID test sets. Together, these two settings provide a comprehensive and realistic evaluation for continual learning of MLLMs.

Further, we design a novel method to build an efficient, lifelong-evolving MLLM. For plasticity preservation, we employ domain or ability-specific Low-Rank Adaptation (LoRA) modules (Hu et al., 2021) that maintain parameter isolation across sequentially arriving tasks, enabling comprehensive acquisition of new knowledge while preventing catastrophic interference through explicit architectural decoupling. Concurrently, to enhance parameter selection accuracy in complex multimodal scenarios, we devise a multimodal routing mechanism that leverages the model's intrinsic multimodal understanding capabilities to automatically align input patterns with optimal task parameters. This strategy effectively transforms the MLLM's knowledge into an explicit expert selector.

In summary, our main contributions are as follows:

- We establish a novel benchmark for CL of MLLMs, with practical domain and ability continual learning settings, focusing on both IID and non-IID evaluation.

- We propose a simple yet effective method with domain or ability-specific low-rank adaptation and large multimodal model-based parameter selection.

- Experiments show that our method achieves impressive results on both domain and ability settings of the MLLM-CL benchmark, significantly outperforming existing approaches.

## 2 RELATED WORK

**Continual Learning.** Researchers have developed primarily four main strategies for continual learning: rehearsal-based methods (Lavda et al., 2018; Buzzega et al., 2020), regularization-based methods (Kirkpatrick et al., 2017; Li & Hoiem, 2017), structure-based methods (Mallya et al., 2018;

Douillard et al., 2022), and prompt-based methods (Wang et al., 2022; Smith et al., 2023). CL in large language models has recently gained much attention (Wu et al., 2024; Shi et al., 2024a). According to the training stages, we can divide them into continual pre-training (Jang et al., 2022; Cossu et al., 2024), continual instruction tuning (Razdaibiedina et al., 2023; Zan et al., 2022; Yin et al., 2022; Wang et al., 2023a), and continual alignment (Zhang et al., 2024a; Suhr & Artzi, 2024). However, few studies focus on continual learning of MLLMs (Chen et al., 2024a; Zeng et al., 2024; Cao et al., 2024; Guo et al., 2025a;c). These prior attempts establish benchmarks with a simple *dataset incremental setting* where training and test sets are distributed independently and identically. Some works focus on conducting continuous instruction tuning directly from the model after the pretraining process (Huai et al., 2025; He et al., 2023). While these efforts have advanced the development of continual learning for MLLMs to some extent, they exhibit an apparent gap with the real-world production environment. Therefore, our work fills this gap and proposes a comprehensive and practical benchmark, including adding domain-specific knowledge and general abilities for CL of MLLM.

**Multimodal Large Language Models.** Advances in MLLMs have demonstrated remarkable capabilities in multimodal understanding, open-ended generation, and instruction following across modalities. Early efforts, such as LLaVA (Liu et al., 2023; 2024a) and Qwen-VL (Bai et al., 2023), use image encoders (Radford et al., 2021) and projectors to transfer multimodal inputs into language embedding space. Recent studies (OpenAI, 2024; Li et al., 2024a; Bai et al., 2025; Fu et al., 2025) expand the ability of MLLM into more modalities, such as video and audio. With the rapid growth of MLLMs, the costs associated with training from scratch have increased dramatically (Li et al., 2024a; Tong et al., 2024; Bai et al., 2025; Chen et al., 2024c). Therefore, adapting MLLMs to dynamic environments by retraining them from scratch becomes expensive and inefficient, creating an imperative demand for continual learning of MLLMs.

**Training-free Adaptation Strategies.** Beyond parameter tuning, recent research has explored training-free mechanisms for model adaptation. In-context learning (ICL) (Alayrac et al., 2022; Dong et al., 2024; Brown et al., 2020) allows MLLMs to adapt to new tasks via few-shot demonstrations, while retrieval-augmented generation (RAG) (Fan et al., 2024; Gao et al., 2023; Lewis et al., 2020) supplements model knowledge by retrieving external data. While these strategies leverage the generalization of base models, they are often constrained by context window limits and increased inference latency due to lengthy prompts (Gao et al., 2023; Fan et al., 2024). Furthermore, training-free methods rely heavily on the model's pre-existing feature space, which may be insufficient for highly specialized domains (*e.g.*, remote sensing or medical diagnosis). In contrast, our continual learning framework effectively internalizes domain-specific patterns into parameters, which results in "muscle memory" (Lum & Conti-Ramsden, 2013) for new abilities and more efficient inference, making it complementary to training-free approaches for building lifelong-evolving MLLMs.

## 3 MLLM-CL BENCHMARK

In this section, we provide the problem formulation and introduce the continual learning benchmark MLLM-CL. Based on the general ability and domain-specific knowledge updated in the instruction tuning stage, we divide our benchmark into domain continual learning and ability continual learning, respectively. In domain continual learning, we desire the model to learn knowledge continually, and the training sets and the test sets are IID. While in ability continual learning, we desire the model to enhance different abilities from the training data and generalize to non-IID test sets.

**Problem Statement.** Continual learning in MLLMs involves sequentially learning a series of multimodal tasks. Let $\mathcal{X}^{\text{img}}$ and $\mathcal{X}^{\text{ins}}$ denote the image and instruction spaces, respectively, and $\mathcal{Y}$ represent the label space for answers composed of $L$ tokens. Given a sequence of datasets $\mathcal{D}_1, \ldots, \mathcal{D}_T$, where each $\mathcal{D}_t = \{(x_{t,i}^{\text{img}}, x_{t,i}^{\text{ins}}, y_{t,i})\}_{i=1}^{N_t}$ contains $N_t$ image-instruction-answer triplets drawn IID from the task-specific distribution $\mathcal{P}_t = \mathcal{X}_t^{\text{img}} \times \mathcal{X}_t^{\text{ins}} \times \mathcal{Y}_t$. Our goal is to continually update a multimodal model on observed data while retaining knowledge from previous tasks. Denote the model by $f$ with parameters $\theta_t$ at stage $t$, the training objective of MLLM is to predict the next token in an autoregressive way:

$$\mathcal{L}_{\text{MLLM}}(\theta_t) = -\sum_{i=1}^{N_t} \sum_{l=1}^{L} \log p_{\theta_t}(y_{t,i}^l | x_{t,i}^{\text{img}}, x_{t,i}^{\text{ins}}, y_{t,i}^{<l}). \quad (1)$$

Table 1: Statistics of the training datasets and test datasets for domain continual learning and ability continual learning. In domain continual learning, "RS" stands for remote sensing, "Med" is medical, "AD" is autonomous driving, "Sci" stands for science, and "Fin" means finance. In ability continual learning, "M & L" stands for math & logic. "VP" means visual perception.

| Task | Train Dataset | Test Dataset | Train Number | Test Number |
|---|---|---|---|---|
| *Domain Continual Learning* | | | | |
| RS | RSVQA | RSVQA | 60k | 10k |
| Med | PathVQA | PathVQA | 22.8k | 9.8k |
| AD | DriveLM | DriveLM | 60k | 10k |
| Sci | AI2D, SciVerse | AI2D, SciVerse | 33.4k | 8.2k |
| | MapQA, TQA | MapQA, TQA | (12.4k, 0.9k, 9.6k, 7.8k) | (3.1k, 0.2k, 2.4k, 1.9k) |
| Fin | StockQA | StockQA | 60k | 10k |
| *Ability Continual Learning* | | | | |
| OCR | Monkey | OCRBench | 128.1k | 1k |
| M & L | MathV360K, MAVIS | MathVista | 526.1k | 1k |
| VP | CLEVR, TallyQA | CV-Bench | 119.9k | 0.8k |
| GUI Agent | ScreenQA, MultiUI Screen2Words | MMTBench | 147.3k | 0.8k |

At inference time, given an image-instruction pair $(x^{\text{img}}, x^{\text{ins}})$ drawn from all learned task distributions $\{\mathcal{P}_j\}_{j=1}^t$, the model generates tokens autoregressively, *i.e.*, the $l$-th output token is $\hat{y}^l = \arg\max_{v \in \mathcal{V}} p_\theta(v|x^{\text{img}}, x^{\text{text}}, \hat{y}^{<l})$. The above describes a typical IID scenario (*e.g.*, domain-specific evaluation) where training and test data belong to $\{\mathcal{P}_j\}_{j=1}^t$. In other cases, the model can encounter various out-of-distribution inputs $\{\mathcal{P}_{j,\text{non-iid}}\}_{j=1}^t \neq \{\mathcal{P}_j\}_{j=1}^t$ (*e.g.*, ability evaluation where the input images and instruction style can be diverse), and is supposed to handle non-IID scenarios.

To address the need for a more comprehensive evaluation, we introduce two distinct yet complementary settings: **domain continual learning (DCL)** and **ability continual learning (ACL)**. DCL evaluates the model's capacity to acquire and retain specialized knowledge within specific domains (*e.g.*, Medical, Finance) under an independently and identically distributed (IID) setting. Here, training and testing data are drawn from the same underlying distribution, directly measuring the model's retention of explicitly taught knowledge. In contrast, ACL assesses the model's ability to generalize fundamental skills (*e.g.*, OCR, Math) to novel scenarios. This is achieved through a non-IID evaluation where the model is trained on one dataset but tested on another, thereby measuring its plasticity and true generalization capabilities. Together, these two settings provide a more holistic and realistic testbed for an MLLM's lifelong learning prowess.

**Domain Continual Learning (DCL).** Continually adding domain knowledge is crucial for constructing a powerful MLLM. To achieve this goal, we propose domain continual learning and choose five mainstream and common domains: remote sensing, medical, science, autonomous driving, and finance. Specifically, we choose RSVQA (Lobry et al., 2020), PathVQA (He et al., 2020), DriveLM (Sima et al., 2023), FinVis (Wang et al., 2023b), AI2D (Kembhavi et al., 2016), SciVerse (Guo et al., 2025e), MapQA (Chang et al., 2022) and TQA (Kembhavi et al., 2017). However, FinVis is a caption dataset in Chinese, which may result in a language gap and is not convenient for evaluation. Therefore, we regenerate the SFT and test data as multiple-choice questions and yes-or-no questions using a *questioner-inspector* data pipeline. Fig. 2 shows the overall data pipeline. We use two agents, a QA generator and an inspector. Considering the varying task difficulties, we use Qwen2.5-VL-72b (Bai et al., 2025) to generate multiple choice QA pairs and Qwen2.5-VL-7b to generate Y/N QA pairs. For the inspector, we use

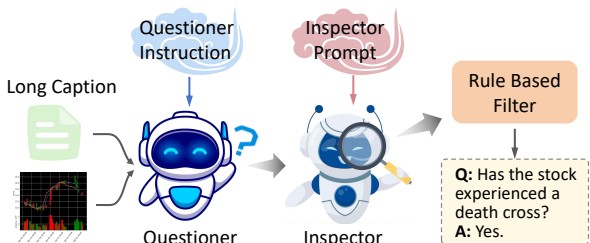

Figure 2: The questioner-inspector data pipeline for generating StockQA instruction tuning dataset.

Qwen2.5-VL-7b to check the correctness of each QA pair. After initial inspection, rule-based formatting is applied to generate the final dataset, named StockQA. All experiments are conducted using

```
You are a helpful assistant router. There are five expert models, each specializing in one of the
following domains: finance (stock), science, medical imaging, autonomous driving, and remote sensing.

Your task is to select the most suitable model based on the provided visual content, user question,
and model descriptions. Consider the expertise of each model carefully and select the one best
equipped to handle the given question.

Important Instructions:
• Respond only with the letter (A,B,C,D,E) corresponding to the most suitable model.
• Do not attempt to answer the user's question directly.

Model Pool:

• A: A financial expert specializing in stock market analysis using candlestick charts. This model
  excels at trend prediction and technical indicator analysis.
• B: A science expert with proficiency in biology, map interpretation, physics, and chemistry.
• C: A medical imaging expert, primarily focused on pathology, including cell sections and natural
  images of medical conditions.
• D: An autonomous driving expert specializing in ego-view scene understanding, including
  coordinate prediction and action planning and other driving-related tasks. The input image is an
  image concatenated by 6 camera views.
• E: A remote sensing expert, adept at analyzing aerial or satellite images. This model excels at
  object counting, presence detection, and area estimation.

Here is the user's question: [User's Question]
```

Figure 3: Prompt of the MLLM-based router selector.

the vllm (Kwon et al., 2023) engine. *Appendix* B provides detailed prompts for each agent, rules for filtering, examples, and statistics of the StockQA dataset. Tab. 1 shows the statistics of the datasets for DCL and Fig. 1 shows some examples. More examples are provided in the *Appendix* G.1.

**Ability Continual Learning (ACL).** As noted, the DCL setting assumes that training and test data are IID. However, this is often not the case in real-world scenarios, a challenge ignored by existing benchmarks (Chen et al., 2024a; Zeng et al., 2024; Guo et al., 2025a; Cao et al., 2024). Therefore, our ACL setting considers the more challenging non-IID scenario. For ACL, we select four fundamental abilities for the MLLM to learn sequentially: OCR, math & logic, visual perception, and GUI agent. In terms of the SFT data, we collect the training data from LLaVA-OneVision (Li et al., 2024a), Monkey (Li et al., 2024b), ScreenQA (Hsiao et al., 2022), Screen2Words (Wang et al., 2021), MultiUI (Liu et al., 2024b), Math-LLaVA (Shi et al., 2024b), MAVIS (Zhang et al., 2024b), CLVER (Johnson et al., 2017) and TallyQA (Acharya et al., 2019) and testing data from OCRBench (Liu et al., 2024d), MathVista (Lu et al., 2024), MMTBench-GUI (Ying et al., 2024) and CV-Bench-Counting (Tong et al., 2024), respectively. Tab. 1 presents the details of the datasets in ACL, and Fig. 1 provides a demonstration. Additional examples can be found in the *Appendix* G.1.

## 4 THE PROPOSED METHOD: MR-LORA

### 4.1 TRAINING: EXPERT LEARNING WITHOUT TASK CONFLICT

**Learning Low-Rank Expert without Task Conflict.** In traditional continual learning, particularly class-incremental learning, the model for learning a new task is typically initialized with parameters from the previous task to facilitate knowledge transfer, and then various regularization constraints are incorporated to mitigate catastrophic forgetting. Therefore, a natural question arises: Is this paradigm suitable for continual learning in MLLMs? Some studies (Wei et al., 2025; Yang et al., 2024) have revealed that data interference widely exists in the training of MLLMs. We empirically investigate the task

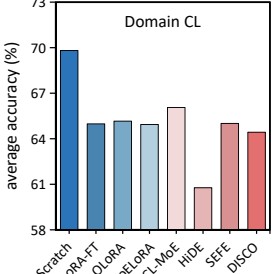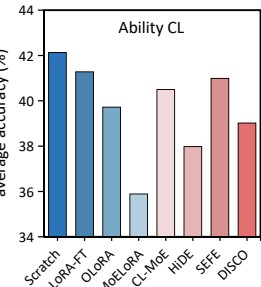

Figure 4: Comparison of new task performance (LLaVA-based) on both domain and ability CL.

conflict problem of domain and ability continual learning by comparing the average new task perfor-

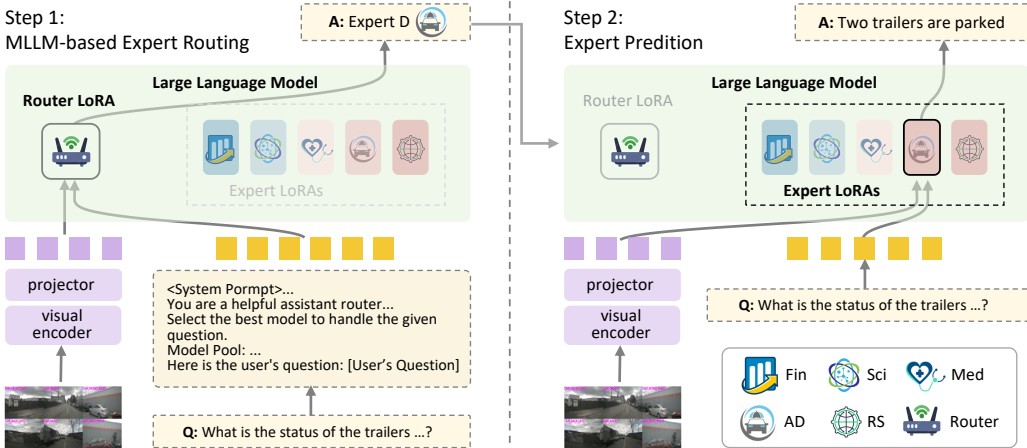

Figure 5: Overall framework of our MR-LoRA.

mance. The results in Fig. 4 yield the following observation: (1) Initializing with weights from prior tasks (*e.g.*, LoRA-FT, MoELoRA (Chen et al., 2024a)) reduces model plasticity, leading to worse performance than learning each task individually with randomly initialized LoRA (*i.e.*, scratch). (2) Regularization (*e.g.*, O-LoRA (Wang et al., 2023a), SEFE (Chen et al., 2025)) or parameter-sharing-based methods (*e.g.*, CL-MoE (Huai et al., 2025), HiDE (Guo et al., 2025a)) also suffer from loss of plasticity when learning new tasks. (3) The task conflict in DCL is more severe than that in ACL, which is reasonable because the domain gap in DCL (*e.g.*, autonomous driving vs. science) is often larger than that in ACL (OCR vs. Math). Based on the above analysis, we propose initializing a fresh LoRA (Hu et al., 2021) module from scratch for each task to circumvent inter-task conflicts when learning new domains. Compared to the original parameters of the large model, LoRA introduces minimal parameters, enabling domain-specific adaptation via lightweight, task-exclusive adapters.

**Few-shot Router Tuning.** In our framework, we tune a low-rank expert for each domain or capability, and dynamically select the most appropriate expert at inference time. While existing selection strategies (Zeng et al., 2024; Guo et al., 2025a) rely on simple similarity measures, *e.g.*, computing cosine similarity between task prototypes and sample features in the embedding space, multimodal scenarios involve more complex inputs. Therefore, we propose leveraging the MLLM's intrinsic capability to process complex multimodal inputs by tuning an MLLM-based selection router. This router identifies the corresponding expert for each input. Specifically, for each task, we collect a few-shot set $\mathcal{M}_t = \{(x_{t,i}^{\text{img}}, x_{t,i}^{\text{ins}})\}_{i=1}^m$, where $m \ll N_t$ (we set $m = 20$ in all experiments). After each continual learning phase, the accumulated few-shot data $\{\mathcal{M}_j\}_{j=1}^t$ and expert model descriptions are transformed into structured instructions. We adopt a *generative* style to select the most suitable expert and tune the MLLM using a router LoRA via autoregressive loss (Liu et al., 2024a). An illustration of the router selection prompt for domain continual learning is provided in Fig. 3.

## 4.2 INFERENCE: ROUTER SELECTION WITH MLLM

**Framework of MR-LoRA.** During inference, with expert learning and router selection, the overall framework of the proposed method is illustrated in Fig. 5. Our MR-LoRA performs two-stage inference for a given multimodal input, consisting of a routing phase followed by a prediction phase. In the first stage, the expert selection router is performed to select a domain or ability-specific expert. Then, the selected expert is combined with the pre-trained backbone to output the final response. On the one hand, by decoupling the learning of different domains or abilities, we avoid potential distribution conflict and can learn a good expert for a given task. On the other hand, the proposed router selection strategy largely explores the advantages of MLLMs to improve the flexibility and accuracy of expert selection, ensuring promising final prediction performance during continual learning. The proposed MLLM-based routing mechanism offers notable advantages: (1) The MLLM's strong multimodal understanding capacity ensures robust expert selection performance on complex multimodal inputs. (2) The selection router is parameter-efficient and learned with few-shot unlabeled image-question pairs, allowing on-the-fly adaptation.

## 5 EXPERIMENTS

### 5.1 EXPERIMENTAL SETUP

**Model and Compared Methods.** We conduct experiments on LLaVA-v1.5-7b (Liu et al., 2023) and InternVL (Chen et al., 2024d) to continually increase the domain-specific knowledge and abilities in our MLLM-CL benchmark, respectively. All the continual learning experiments start from the instruct models, *i.e.*, LLaVA-v1.5-7b and InternVL-Chat-V1.0. For the task sequence in domain continual learning, we choose a random order of remote sensing→medical→autonomous driving→science→finance. For ability continual learning, we set the task sequence as OCR→math & logic→visual perception→GUI agent. We choose CL-MoE (Huai et al., 2025), SEFE (Chen et al., 2025), DISCO (Guo et al., 2025b), O-LoRA (Wang et al., 2023a), HiDE (Guo et al., 2025a), MoELoRA (Chen et al., 2024a), ModalPrompt (Zeng et al., 2024), L2P (Wang et al., 2022), and LoRA (Hu et al., 2021) as baselines using the MCITlib (Guo et al., 2025d) to show the effectiveness of our proposed method in the two settings of MLLM-CL. For baselines utilizing replay (denoted by *), we adopt a data merging strategy where replay samples from previous tasks are added directly to the current task's training set. We also report the zero-shot and oracle performance for each setting. Oracle performance is achieved by training an individual LoRA from the base model and subsequently evaluating its performance.

**Evaluation Metric.** We report the following standard metrics for CL (Guo et al., 2025a; Chen et al., 2025): Last accuracy is the accuracy of all seen tasks after learning the last task. Mean Finetune Accuracy (MFT) measures the average accuracy achieved on each task immediately after it is learned, serving as an upper bound that reflects the model's performance in the absence of forgetting. Mean Final Accuracy (MFN) computes the average accuracy over all tasks after completing the full incremental training process, representing the model's overall retained performance. Mean Average Accuracy (MAA) calculates the mean of average accuracies on all learned tasks after each training step, offering a holistic view of performance throughout the CL process. Backward Transfer (BWT) captures the change in accuracy for each task by comparing its final accuracy with that immediately after it was learned, quantifying the extent of forgetting. The detailed calculation of each metric is shown in Fig. 6:

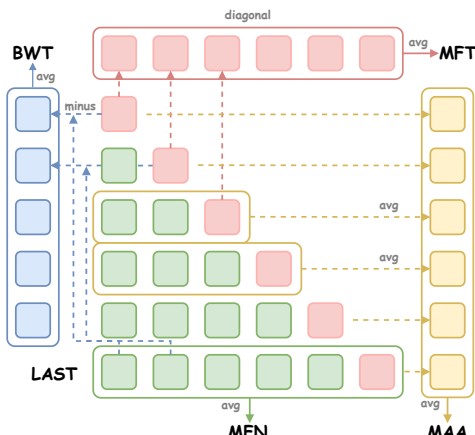

Figure 6: Illustration of metrics. The matrix plots training tasks (rows) against evaluation tasks (columns).

The diagonal represents the model's accuracy immediately after learning a specific task (basis for MFT). The bottom row shows the final performance on all tasks after the complete training sequence (basis for MFN). BWT is measured by comparing a task's initial score on the diagonal against its final score in the bottom row to quantify forgetting. Finally, MAA aggregates the entire lower triangle to track the overall model's holistic performance throughout the continual learning lifecycle.

### 5.2 RESULTS AND ANALYSIS

**Domain Continual Learning.** As demonstrated in Tab. 2 (LLaVA-based) and Tab. 4 (InternVL-based), our proposed MR-LoRA method achieves state-of-the-art performance on the DCL setting, showcasing its exceptional ability to acquire new domain knowledge while preserving previously learned capabilities. The performance of MR-LoRA highlights several key advantages: (1) *Approaching oracle performance:* Our method's final accuracy on all individual tasks nearly matches the "Oracle" performance. For instance, in Tab. 2, the final accuracies of MR-LoRA across the five domains are almost identical to the Oracle scores. This indicates that our MLLM-based router can select the most appropriate expert module for each input sample with high precision, allowing the overall performance to approach the theoretical upper bound of a perfect selection mechanism. (2) *Superiority over existing baselines:* In contrast, other baseline methods exhibit significant performance degradation. Parameter-sharing and regularization methods like LoRA-FT and O-LoRA

Table 2: Results for LLaVA-based domain continual learning in MLLM-CL benchmark. * denotes the original method with replay data.

| Method | RS | Med | AD | Sci | Fin | MFT↑ | MFN↑ | MAA↑ | BWT↑ |
|---|---|---|---|---|---|---|---|---|---|
| Zeroshot | 32.29 | 28.28 | 15.59 | 35.55 | 62.56 | 34.85 | - | - | - |
| Oracle | 81.06 | 65.83 | 54.17 | 56.86 | 91.14 | 69.81 | - | - | - |
| LoRA-FT (Hu et al., 2021) | 69.65 | 41.59 | 25.43 | 40.88 | 87.45 | 64.98 | 53.00 | 61.13 | -14.97 |
| LoRA-FT* (Hu et al., 2021) | 76.54 | 50.27 | 43.01 | 43.32 | 89.85 | 66.32 | 60.60 | 64.72 | -7.15 |
| O-LoRA (Wang et al., 2023a) | 74.64 | 44.42 | 30.02 | 41.47 | 87.15 | 65.16 | 55.54 | 62.12 | -12.03 |
| O-LoRA* (Wang et al., 2023a) | 76.94 | 41.17 | 34.18 | 39.61 | 83.22 | 60.49 | 55.02 | 60.73 | -6.83 |
| MoELoRA (Chen et al., 2024a) | 77.54 | 41.85 | 27.62 | 40.13 | 86.75 | 64.94 | 54.78 | 61.76 | -12.70 |
| MoELoRA* (Chen et al., 2024a) | 77.63 | 49.54 | 39.08 | 41.04 | 89.21 | 66.24 | 59.30 | 64.81 | -8.68 |
| CL-MoE (Huai et al., 2025) | 71.34 | 46.84 | 26.33 | 41.17 | 88.74 | 66.06 | 54.88 | 61.79 | -13.96 |
| CL-MoE* (Huai et al., 2025) | 76.58 | 52.31 | 39.65 | 45.64 | 90.21 | 66.65 | 60.88 | 64.95 | -7.22 |
| HiDe (Guo et al., 2025a) | 74.31 | 48.95 | 33.21 | 38.54 | 81.55 | 60.77 | 55.31 | 60.68 | -6.82 |
| HiDe* (Guo et al., 2025a) | 74.80 | 42.29 | 34.03 | 38.01 | 79.22 | 60.83 | 53.67 | 61.81 | -8.95 |
| SEFE (Chen et al., 2025) | 77.26 | 50.37 | 37.21 | 40.87 | 86.82 | 65.01 | 58.51 | 63.63 | -8.13 |
| SEFE* (Chen et al., 2025) | 78.43 | 52.85 | 46.21 | 47.76 | 89.33 | 66.89 | 62.92 | 66.51 | -4.97 |
| DISCO (Guo et al., 2025b) | 76.03 | 45.20 | 43.79 | 42.33 | 88.95 | 64.43 | 59.26 | 63.35 | -6.46 |
| DISCO* (Guo et al., 2025b) | 77.78 | 46.25 | 50.45 | 49.51 | 89.71 | 65.27 | 62.74 | 64.92 | -3.17 |
| ModalPrompt (Zeng et al., 2024) | 53.63 | 45.68 | 40.77 | 41.81 | 87.82 | 53.87 | 53.94 | 49.67 | **0.09** |
| L2P (Wang et al., 2022) | 63.82 | 34.63 | 22.96 | 38.58 | **92.98** | 66.96 | 50.59 | 59.23 | -20.46 |
| MR-LoRA (Ours) | **80.87** | **65.32** | **54.12** | **56.71** | 91.12 | **69.64** | **69.63** | **71.06** | -0.01 |

Table 3: Results for LLaVA-based ability continual learning in MLLM-CL benchmark.

| Method | OCR | M&L | VP | GUI Agent | MFT↑ | MFN↑ | MAA↑ | BWT↑ |
|---|---|---|---|---|---|---|---|---|
| Zeroshot | 31.20 | 30.20 | 60.79 | 10.00 | 33.05 | - | - | - |
| Oracle | 33.60 | 36.50 | 65.10 | 32.50 | 41.93 | - | - | - |
| LoRA-FT (Hu et al., 2021) | 23.60 | 33.70 | 55.84 | 32.50 | 41.28 | 36.41 | 36.58 | -6.49 |
| LoRA-FT* (Hu et al., 2021) | 21.80 | 32.70 | 58.38 | 28.75 | 40.32 | 35.41 | 36.32 | -6.55 |
| O-LoRA (Wang et al., 2023a) | 29.60 | 32.90 | 52.41 | **33.75** | 39.72 | 37.16 | 35.42 | -3.41 |
| O-LoRA* (Wang et al., 2023a) | 29.60 | 31.30 | 60.79 | 27.50 | 39.96 | 37.30 | 36.34 | -3.55 |
| MoELoRA (Chen et al., 2024a) | 26.70 | 32.80 | 56.85 | 27.22 | 39.45 | 35.89 | 36.07 | -4.75 |
| MoELoRA* (Chen et al., 2024a) | 19.80 | 32.20 | 54.19 | 30.00 | 40.35 | 34.05 | 35.39 | -8.41 |
| CL-MoE (Huai et al., 2025) | 19.90 | 32.70 | 53.43 | 30.69 | 40.50 | 34.18 | 35.65 | -8.43 |
| CL-MoE* (Huai et al., 2025) | 25.40 | 31.80 | 60.91 | 30.00 | 41.22 | 37.03 | 37.28 | -5.59 |
| HiDe (Guo et al., 2025a) | 24.60 | 32.10 | 46.32 | 28.75 | 37.98 | 32.94 | 34.60 | -6.72 |
| HiDe* (Guo et al., 2025a) | 24.60 | 28.40 | 30.71 | 23.75 | 36.84 | 26.86 | 33.54 | -13.30 |
| SEFE (Chen et al., 2025) | 26.00 | 33.40 | 57.74 | **33.75** | 40.98 | 37.72 | 36.59 | -4.35 |
| SEFE* (Chen et al., 2025) | 25.60 | 34.80 | 57.61 | 31.39 | **42.25** | 37.35 | 37.93 | -6.53 |
| DISCO (Guo et al., 2025b) | 32.90 | 33.10 | 60.15 | 30.14 | 39.02 | 39.07 | 36.57 | 0.07 |
| DISCO* (Guo et al., 2025b) | **34.20** | 35.00 | 61.55 | 27.50 | 40.14 | 39.56 | 37.85 | -0.77 |
| Modalprompt (Zeng et al., 2024) | 31.80 | 32.50 | 60.53 | 10.00 | 33.61 | 33.71 | 34.16 | **0.13** |
| L2P (Wang et al., 2022) | 25.10 | 32.00 | 48.22 | 16.25 | 35.48 | 30.39 | 33.78 | -6.78 |
| MR-LoRA (Ours) | 33.70 | **36.20** | **65.10** | 32.50 | 41.89 | **41.88** | **38.86** | -0.02 |

suffer from severe forgetting, as evidenced by their deeply negative BWT scores (*e.g.*, -14.97 for LoRA-FT on LLaVA). This empirically confirms our hypothesis in Sec. 4.1 regarding the severe task conflict among heterogeneous domains, where shared parameters compromise existing abilities while learning new ones. Although replay-based methods (marked with *) alleviate forgetting by rehearsing old data, their performance remains far inferior to MR-LoRA. Even more advanced baselines like DISCO* and SEFE* still show a significant gap compared to ours.

**Ability Continual Learning.** The effectiveness of our proposed method in the more challenging ACL setting is demonstrated in Tabs. 3 and 5. This setting evaluates the model's capacity to acquire fundamental new skills and generalize to non-IID test sets. Firstly, we observe that most baselines suffer from severe catastrophic forgetting, revealing a critical weakness in existing CL approaches when faced with real-world, practical non-IID scenarios. In contrast, our MR-LoRA significantly outperforms all baseline methods and successfully improves performance across all four abilities by isolating abilities into dedicated expert modules and leveraging an intelligent MLLM-based router.

**Compensatory Contribution from Non-Designated Experts.** A phenomenon we term "compensatory contribution from non-designated experts" reveals the fundamental distinction between our two learning settings. In DCL, which operates under an IID evaluation, such contributions are rare because

Table 4: Results for InternVL-based domain continual learning in MLLM-CL benchmark. * denotes the original method with replay data.

| Method | RS | Med | AD | Sci | Fin | MFT↑ | MFN↑ | MAA↑ | BWT↑ |
|---|---|---|---|---|---|---|---|---|---|
| Zeroshot | 31.16 | 29.81 | 14.06 | 33.93 | 64.32 | 34.66 | - | - | - |
| Oracle | 81.49 | 66.42 | 54.56 | 54.48 | 91.24 | 69.64 | - | - | - |
| LoRA-FT (Hu et al., 2021) | 69.93 | 52.17 | 33.04 | 42.67 | 91.07 | 69.06 | 57.78 | 65.22 | -14.11 |
| LoRA-FT* (Hu et al., 2021) | 77.06 | 47.55 | 42.67 | 43.31 | **91.44** | 69.43 | 60.41 | 67.45 | -11.28 |
| MoELoRA (Chen et al., 2024a) | 69.90 | 52.08 | 33.17 | 42.19 | 90.58 | 68.83 | 57.58 | 65.97 | -14.06 |
| MoELoRA* (Chen et al., 2024a) | 76.74 | 52.65 | 38.81 | 42.15 | 89.84 | 67.90 | 60.04 | 66.01 | -9.83 |
| HiDe (Guo et al., 2025a) | 75.40 | 57.66 | 36.73 | 41.48 | 88.59 | 65.26 | 59.97 | 65.94 | -6.60 |
| HiDe* (Guo et al., 2025a) | 53.17 | 52.61 | 40.85 | 47.04 | 89.17 | 64.20 | 56.57 | 61.06 | -9.54 |
| DISCO (Guo et al., 2025b) | 75.12 | 50.69 | 52.41 | 50.67 | 90.86 | 68.85 | 63.95 | 68.14 | -6.12 |
| DISCO* (Guo et al., 2025b) | 77.90 | 47.50 | 49.13 | 49.37 | 90.92 | 68.55 | 62.96 | 67.81 | -6.98 |
| SEFE (Chen et al., 2025) | 78.21 | 57.59 | 51.45 | 44.65 | 91.37 | **69.55** | 64.65 | 68.84 | -6.12 |
| CL-MoE (Huai et al., 2025) | 78.12 | 52.51 | 35.53 | 42.69 | 91.24 | 69.22 | 60.02 | 67.60 | -11.51 |
| ModalPrompt (Zeng et al., 2024) | 50.20 | 30.95 | 14.11 | 33.91 | 65.45 | 38.93 | 38.92 | 38.70 | **-0.01** |
| O-LoRA (Wang et al., 2023a) | 74.48 | 54.16 | 39.60 | 48.30 | 88.54 | 65.51 | 61.02 | 65.83 | -5.62 |
| MR-LoRA (Ours) | **81.48** | **65.80** | **54.56** | **54.40** | 91.07 | 69.51 | **69.46** | **71.27** | -0.06 |

Table 5: Results for InternVL-based ability continual learning in MLLM-CL benchmark.

| Method | OCR | M&L | VP | GUI Agent | MFT↑ | MFN↑ | MAA↑ | BWT↑ |
|---|---|---|---|---|---|---|---|---|
| Zeroshot | 30.00 | 31.20 | 56.09 | 2.50 | 29.95 | - | - | - |
| Oracle | 32.20 | 33.40 | 67.77 | 33.75 | 41.78 | - | - | - |
| LoRA-FT (Hu et al., 2021) | 21.40 | 32.80 | 60.28 | 29.86 | 40.84 | 36.08 | 36.38 | -6.35 |
| LoRA-FT* (Hu et al., 2021) | 26.30 | 34.20 | 62.56 | 31.25 | 41.63 | 38.58 | 37.38 | -4.07 |
| O-LoRA (Wang et al., 2023a) | 25.50 | 32.30 | 64.59 | 24.44 | 38.64 | 36.71 | 36.05 | -2.57 |
| O-LoRA* (Wang et al., 2023a) | 21.70 | 31.10 | 59.77 | 31.25 | 41.38 | 35.96 | 36.49 | -7.23 |
| MoELoRA (Chen et al., 2024a) | 17.20 | 32.70 | 55.33 | 32.50 | 41.41 | 34.43 | 35.36 | -9.30 |
| MoELoRA* (Chen et al., 2024a) | 13.90 | 29.70 | 54.95 | 32.50 | 41.91 | 32.76 | 35.66 | -12.20 |
| HiDe (Guo et al., 2025a) | 17.70 | 33.00 | 41.12 | 20.28 | 37.27 | 28.02 | 33.25 | -12.33 |
| HiDe* (Guo et al., 2025a) | 25.30 | 29.20 | 42.13 | 20.28 | 35.93 | 29.23 | 33.39 | -8.93 |
| DISCO (Guo et al., 2025b) | 30.60 | 33.10 | 65.36 | 27.50 | 39.21 | 39.14 | 36.73 | -0.10 |
| DISCO* (Guo et al., 2025b) | 32.30 | 32.30 | 64.97 | 30.14 | 40.46 | 39.93 | 37.63 | -0.71 |
| SEFE (Chen et al., 2025) | 19.00 | 32.20 | 62.18 | 31.94 | 41.45 | 36.33 | 36.29 | -6.83 |
| CL-MoE (Huai et al., 2025) | 25.10 | 31.90 | 60.91 | 27.50 | 40.15 | 36.35 | 35.75 | -5.06 |
| ModalPrompt (Zeng et al., 2024) | 25.90 | 31.60 | 55.58 | 11.39 | 32.55 | 31.12 | 31.44 | -1.90 |
| MR-LoRA (Ours) | **33.00** | **35.70** | **67.51** | **33.75** | **42.56** | **42.49** | **38.85** | **-0.09** |

the designated domain expert is typically optimal. However, they remain possible in real-world scenarios with ambiguous semantic boundaries, highlighting our router's flexibility. In contrast, the ACL setting, characterized by its non-IID evaluation, fosters the development of shared foundational skills. For instance, the math expert implicitly acquires robust OCR capabilities to parse equations. Our MLLM-based router can leverage this, acting as a dynamic ensemble to select the math expert for a challenging OCR sample where its latent digital-reading ability surpasses that of the designated OCR expert (Fig. 7). This mechanism of beneficial deviation explains why MR-LoRA's performance can even surpass the Oracle baseline, as empirically observed in the OCR task. Consequently, this behavior underscores the router's nuanced, instance-level reasoning capabilities that extend beyond simple task labels, validating the distinct evaluative objectives of the DCL and ACL frameworks.

**Rank of Expert LoRA.** From the results in Tab. 6, we find that our method performs well even at very low ranks (*e.g.*, 8), demonstrating its parameter efficiency. This indicates that even if the number of tasks to be learned is large, our method can still achieve a good performance with only a small increase in parameters. Besides, as the expert rank increases, performance can be improved slightly because of more trainable parameters.

Table 6: Ablation study of LoRA rank for each expert LoRA (LLaVA, DCL, last accuracy).

| Rank | RS (%) | Med (%) | AD (%) | Sci (%) | Fin (%) |
|---|---|---|---|---|---|
| 8 | 80.96 | 64.64 | 54.00 | 55.44 | 90.75 |
| 16 | 80.92 | 65.11 | 53.98 | 55.90 | 91.02 |
| 32 | 80.87 | 65.32 | 54.12 | 56.71 | 91.12 |
| 64 | 81.18 | 66.07 | 54.31 | 56.90 | 91.60 |
| 128 | 81.14 | 66.49 | 54.00 | 57.63 | 91.44 |

**Router Accuracy.** We ablate the number of samples for routing data and report the router selection accuracy and the last accuracy in domain and ability continual learning. The results are shown in Tabs. 7 and 8. In DCL, we find that our method can achieve an excellent performance (almost 100%

Figure 7: Examples demonstrating that the selected expert handles certain questions better than the original expert in DCL and ACL. MLLM-enhanced router selects the most appropriate experts.

Table 7: Router accuracy under different amount of router data in *domain continual learning*. The left part is the router selection accuracy and the right part is task accuracy after learning the last task.

| # Replay Samples | Router Accuracy (%) | | | | | Last Accuracy (%) | | | | |
|---|---|---|---|---|---|---|---|---|---|---|
| | RS | Med | AD | Sci | Fin | RS | Med | AD | Sci | Fin |
| 100 | 99.96 | 99.16 | 99.98 | 98.44 | 99.99 | 81.04 | 65.61 | 54.16 | 56.77 | 91.13 |
| 50 | 99.85 | 98.69 | 99.94 | 98.82 | 100.00 | 81.00 | 65.53 | 54.14 | 56.76 | 91.14 |
| 30 | 99.62 | 98.89 | 100.00 | 96.90 | 99.86 | 80.92 | 65.53 | 54.17 | 56.59 | 91.08 |
| 20 | 99.52 | 97.87 | 99.89 | 98.40 | 99.80 | 80.87 | 65.32 | 54.12 | 56.71 | 91.12 |
| 10 | 99.93 | 98.24 | 99.93 | 97.75 | 99.40 | 81.04 | 65.40 | 54.16 | 56.63 | 91.01 |

Table 8: Router accuracy under different amount of replay data in *ability continual learning*.

| # Replay Samples | Router Accuracy (%) | | | | Last Accuracy (%) | | | |
|---|---|---|---|---|---|---|---|---|
| | OCR | M&L | VP | GUI Agent | OCR | M&L | VP | GUI Agent |
| 100 | 72.10 | 94.60 | 99.87 | 100.00 | 32.80 | 36.30 | 65.10 | 32.50 |
| 50 | 65.30 | 83.90 | 99.11 | 100.00 | 32.70 | 36.10 | 64.85 | 32.50 |
| 30 | 53.60 | 90.90 | 97.21 | 98.38 | 33.80 | 36.70 | 64.85 | 32.50 |
| 20 | 51.40 | 86.00 | 100.00 | 100.00 | 33.70 | 36.20 | 65.10 | 32.50 |
| 10 | 81.90 | 76.30 | 100.00 | 100.00 | 32.80 | 35.80 | 65.10 | 32.50 |

selection accuracy) using only 20 samples to train the router, which means our method closes the gap of training each task individually. Note that the number of samples we used is much smaller than the number of training samples (60k). Besides, with more sampling data, the router selection accuracy improves and the performance of MR-LoRA slightly increases. In ACL, the performance of MR-LoRA achieves satisfactory performance when the shot of router tuning is 10. It is interesting that the router accuracy of the OCR task is around 50%, but our method can achieve a comparable, or even better performance compared with directly finetuning an OCR LoRA expert (33.60%). This means MR-LoRA uses other experts to solve the OCR task, and these experts perform well on these test samples. It is reasonable that OCR is a basic and fundamental ability that the math and GUI Agent experts are also able to extract equations and web texts from the images.

## 6 CONCLUSION

In this paper, we first propose MLLM-CL benchmark, a novel benchmark including domain continual learning and ability continual learning. In domain continual learning, we select five specific domains (remote sensing, medical, science, autonomous driving, and finance) and focus on IID evaluation. In ability continual learning, we consider a more practical setting where the training and test sets are non-IID. We select four common and fundamental abilities for MLLM to learning sequentially: OCR, math & logic, visual perception, and GUI agent. To solve the two settings in the MLLM-CL benchmark, we first analyze the task conflict between different tasks and then propose an MLLM enhanced router selection method MR-LoRA. Comprehensive experiments and analyses validate the necessity of our MLLM-CL benchmark and show the effectiveness and efficiency of our proposed method. We believe that our carefully designed benchmark and MR-LoRA can serve as a foundation for continual learning in multimodal large language models and will introduce an innovative and practical direction of continual learning and MLLM to the community.

## ETHICS STATEMENT

Our research is grounded in ethical practices, with particular attention paid to the responsible use of data. This work exclusively employs public, well-established datasets from the MLLM community, and we list all used assets' licenses in Tab. 13. Our use of this data is in accordance with their provided licenses and intended academic purpose.

## REPRODUCIBILITY STATEMENT

To facilitate the reproducibility of our research, we provide comprehensive implementation details in *Appendix A*, including training procedures and hyperparameters. We also report all the result matrices in *Appendix C*. All source code, datasets, and trained models will be publicly released upon the paper's acceptance.

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

# APPENDIX

## A  IMPLEMENTATION DETAILS

In this section, we introduce the implementation details of MR-LoRA and the evaluation details of each task in domain continual learning and ability continual learning.

### A.1  TRAINING DETAILS

**DCL.** Tab. 9 shows the hyperparameters for training the router and expert in domain continual learning. For most configurations, we follow the default setting of LLaVA 1.5 (Liu et al., 2023). To ensure comparable training exposure across datasets of varying sizes, each task is trained for approximately 60,000 instances in DCL. For efficient fine-tuning, a rank of 32 is employed. For all the experiments, we use 8 A100 GPUs, and the training time for each task is around 1 hour.

**ACL.** Tab. 10 shows the hyperparameters for ability continual learning. For ability continual learning, training time is around 20 hours to train all the tasks sequentially.

**Router Training.** For the router training, we train 30 epochs in domain continual learning and ability continual learning; we keep other configurations identical to the experts' except for the learning rate. We use the codebase from MCITlib (Guo et al., 2025d) and LLaVA (Liu et al., 2023).

Table 9: Hyperparameters of MR-LoRA in domain continual learning

|  | Expert Config | | Router Config | |
| --- | --- | --- | --- | --- |
|  | LLaVA | InternVL | LLaVA | InternVL |
| optimizer | AdamW | | AdamW | |
| batch size | 64 | | 64 | |
| lr schedule | cosine decay | | cosine decay | |
| lr warmup ratio | 0.03 | | 0.03 | |
| LoRA rank | 32 | | 32 | |
| DeepSpeed stage | 2 | | 2 | |
| base lr | $1 \times 10^{-4}$ | | $2 \times 10^{-5}$ | $1 \times 10^{-4}$ |
| epoch for RS | 1 | | - | |
| epoch for Med | 3 | | 30 | |
| epoch for AD | 1 | | 30 | |
| epoch for Sci | 2 | | 30 | |
| epoch for Fin | 1 | | 30 | |

Table 10: Hyperparameters of MR-LoRA in ability continual learning

|  | Expert Config | | Router Config | |
| --- | --- | --- | --- | --- |
|  | LLaVA | InternVL | LLaVA | InternVL |
| optimizer | AdamW | | AdamW | |
| batch size | 128 | | 128 | |
| lr schedule | cosine decay | | cosine decay | |
| lr warmup ratio | 0.03 | | 0.03 | |
| LoRA rank | 32 | | 32 | |
| DeepSpeed stage | 2 | | 2 | |
| base lr (OCR) | $5 \times 10^{-5}$ | $2 \times 10^{-4}$ | - | - |
| base lr (M&L,VP,GUI) | $2 \times 10^{-4}$ | $2 \times 10^{-4}$ | $2 \times 10^{-4}$ | $1 \times 10^{-4}$ |
| epoch for OCR | 3 | | - | |
| epoch for Math & Logic | 1 | | 30 | |
| epoch for VP | 1 | | 30 | |
| epoch for GUI Agent | 3 | | 30 | |

Table 11: Complexities and number of trained parameters for each method.

| Method | # of Trained Parameters | # of Saved Parameters | Training Complexity | Inference Complexity & Latency |
|---|---|---|---|---|
| LoRA-FT | Constant (P) | Constant (P) | Low (Baseline) | Negligible (Weights are merged, no overhead) |
| MoELoRA | Constant (N experts + router) | Constant (N experts + router) | Medium | High (Router computation, cannot merge weights) |
| O-LoRA | Constant (P) | Linear(T×P) | High (Orthogonalization loss) | High (Not support merge weights, applies T modules) |
| CL-MoE | Constant (P) | Linear(T×P) | Medium (No router training) | Very High (Similarity search across T tasks, cannot merge) |
| HiDe, DISCO | Constant (P) | Linear(T×P) | Low (Baseline) | Very High (Similarity search across T tasks, cannot merge) |
| SEFE | Constant (P) | Constant (P) | High (Regularization loss) | Negligible (Weights are merged after each task) |
| MR-LoRA | Constant (P) | Linear ((T +1)× P) | Low (Baseline) | Slightly Higher than LoRA-FT (Using KV-cache) |

## A.2 BASELINE DETAILS

To clarify the trade-offs between the compared methods, we have added a summary of their computational complexities and parameter requirements. The primary trade-offs involve how each method manages trainable parameters, saved parameters and the resulting impact on training and inference efficiency. We summarize these aspects in Tab. 11, where T denotes the number of tasks and P represents the parameters in a single LoRA module.

```
You are a helpful assistant router. There are four expert models, each specializing in
one of the following domains: OCR, math & logic, counting, and GUI navigation.
Your task is to select the most suitable model based on the provided visual content, user
question, and model descriptions. Consider the expertise of each model carefully and
select the one best equipped to handle the given question.

Important Instructions:
• Respond only with the letter (A,B,C,D) corresponding to the most suitable model.
• Do not attempt to answer the user's question directly.

Model Pool:
• A: This model excels in OCR tasks, including text extraction, handwriting recognition,
  and document analysis.
• B: This model excels in counting the number of objects in the image. However, it
  struggles to exact text in an image.
• C: This model is an expert in math and logic, including solving equations, geometry,
  and logical reasoning. It is capable of on puzzle test figures, algebraic reasoning
  over functional plots, and scientific reasoning with academic paper figures.
• D: This model is an expert in GUI navigation, including identifying buttons, text
  fields, and other UI elements from screen shots. It is capable of giving coordinates of
  the elements in the image and conduct action on the elements.

Here is the user's question: [User's Question]
```

Figure 8: Prompt for the router in ability continual learning.

## A.3 EVALUATION DETAILS

In domain continual learning, for the financial task, all the questions are MCQ or Y/N questions; we require the prediction to exactly match the ground truth. For autonomous driving, medical, and remote sensing tasks, we consider the prediction to include the ground truth as the correct answer. This serves as the default evaluation method. For science tasks, some test samples are multiple-choice questions (MCQs), and predictions are required to exactly match the ground truth. Certain questions

in MapQA (Chang et al., 2022) require the model to list places; in these cases, we compute the percentage of correct responses. Other science questions are evaluated according to the default method. In ability continual learning, we follow the default setting of the corresponding benchmarks.

## A.4 ROUTER PROMPT FOR MR-LORA

We previously provided our router prompt for DCL in Fig. 3. The prompt for ACL appears in Fig. 8.

```
You are an expert in finance with specialization in stock market analysis. Your task
involves generating a concise, multiple-choice question and answer pair based on a
provided candlestick chart and its corresponding Chinese description.

Guidelines:

1. Question Generation: Formulate a financial question using professional terminology
related to the stock market. Ensure the question is directly based on the information
provided by the candlestick chart. If the questioner thinks the caption does not
correspond to the candlestick chart apparently, the questioner should ignore the caption
and generate questions solely based on the chart.
2. Choices: Provide four distinct options labeled A, B, C, and D. Each option should be
unique and plausible, but only one must be correct. Format the choices as 'A. [Choice_A],
B. [Choice_B], C. [Choice_C], D. [Choice_D]'.
3. Answer: The correct answer should be indicated by its letter (A, B, C, or D) without
any additional text.
4. Output Format: Present the result in the following format: 'Question:[generated
question]Answer:[generated answer]'
5. Ensure the question is concise and clear.
6. The questions and answers must be in English.

Restrictions:
    • Do not predict future trends; base all questions on the given candlestick chart and
      caption.

Please follow these guidelines closely to ensure consistency and clarity in the generated
content. Here is the given caption:

[Caption from FinVis]
```

Figure 9: Prompt for the Questioner to generate MCQ question answer pairs.

## B DETAILS OF STOCKQA DATASET

**Overview.** The StockQA dataset is a multimodal financial dataset concentrated on stock analysis. It is rewritten from the FinVis (Wang et al., 2023b) dataset.

Finvis dataset is a Chinese caption dataset generated by GPT4V (Achiam et al., 2023). All the captions are related to the stock technical indicator analysis. However, the caption form is not convenient for evaluation, and there may be a language gap between this task and other tasks. Therefore, we use a *questioner-inspector* data pipeline with a powerful MLLM Qwen2.5-VL (Bai et al., 2025) to rewrite the caption into MCQ and Y/N question-answer pairs and name it StockQA. When manually checking the inspector process, we find that the inspector *misclassified* some correct question-answer pairs. Nevertheless, it successfully identified erroneous instances, thereby contributing to the overall correctness of the final dataset.

**Prompts for agents.** Figs. 9 and 10 shows the prompt we use for the Questioner to generate Y/N and MCQ question-answer pairs, respectively. Fig. 11 is the prompt we use for the inspector.

**Rules for filtering.** After using an inspector agent to check the correctness and rationality, we employ the following rules to balance the choices of multiple choice questions to mitigate the position bias (Liu et al., 2024c) and format the output.

You are a powerful multimodal model tasked with dual roles: a financial expert and a questioner.

• **Questioner Role**: The questioner receives a candlestick chart along with a caption in Chinese and then asks the expert concise questions in English about different aspects of the stock.
• **Expert Role**: Respond to each question with succinct answers, using no more than 3 words. Your responses should leverage professional financial and stock market terminology, focusing on insights derived from the visual data of the candlestick chart.

**Guidelines**:
    1. Each interaction consists of one 'Q&A' session only.
    2. The question must be a complete sentence (Do not omit any part and be as descriptive as possible) and must be concise and clear, with a maximum length of 20 words.
    3. The caption is a detailed description of the chart. The question can refer to the caption. If the caption does not correspond to the candlestick chart apparently, the questioner should ignore the caption and generate questions solely based on the chart.
    4. Questions should be diverse, covering multiple perspectives such as trend analysis in a specific period, stock price and date at the extreme point, volume indicators, momentum indicators and other reasonable technical indicators of stocks.
    5. The questioner should ask yes/no questions. The answers should be yes or no without further explanations.
    6. Please generate the questions with yes answers and no answers with an equal probability. Do not let one answer dominate.
    7. The questions and answers must be in English.
    8. Please use professional financial and stock market terminology.

**Restrictions**:
    • Do not predict future trends; base all questions on the given candlestick chart and caption.

**Output Format**:
    • Return results in the format: 'Question:[generated question]Answer:[generated answer]'

**Now, generate a relevant question and its corresponding answer based on the provided caption and candlestick chart. Here is the given caption:**

[Caption from FinVis]

Figure 10: Prompt for the Questioner to generate Y/N question answer pairs.

As an expert in financial analysis with the capability to understand complex multimodal inputs, your task is to assess the rationality of a given Question & Answer pair concerning a provided candlestick chart.

1. **Analyze the Question**: Ensure that the question is about the candlestick chart. The information required to answer should be visually extractable from the chart.
2. **Evaluate the Answer**: Verify that the answer correctly interprets the question and accurately reflects the data or trends observable in the candlestick chart.
3. **Judgment**: If both the question is relevant to the chart and the answer is correct based on the chart, respond with "True". In all other cases, respond with "False".

Please provide only one word as your response: either "True" or "False". Do not include any explanations or additional text.

Given Q&A pair for evaluation:

Figure 11: Prompt for the Inspector to check the question answer pairs.

- **Format:** Remove the unnecessary spaces, line breaks, and punctuation to make each question in the same format.

- **Position:** Exchange the choices of multiple choice questions to ensure the right answers of the total datasets are distributed with the same probability.

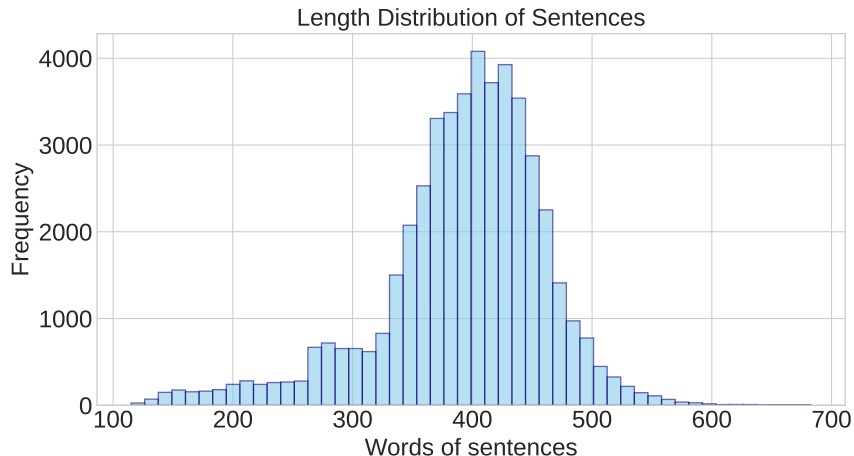

Figure 12: Word length distribution of the StockQA dataset.

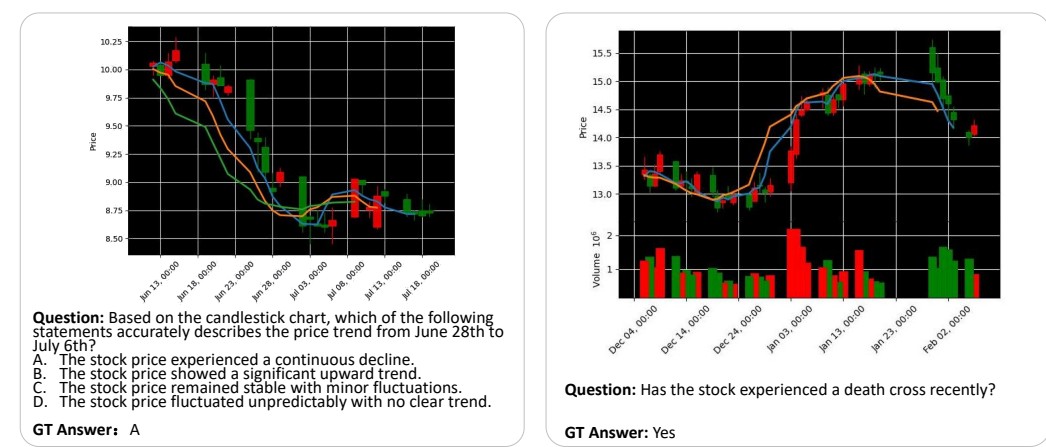

Figure 13: MCQ and Y/N examples in StockQA dataset.

Table 12: Statistics of the StockQA dataset.

| Data | Max Length | Min Length | Average Length | Amount |
|------|------------|------------|----------------|--------|
| MCQ | 683 | 115 | 392.74 | 48k |
| TF | 99 | 21 | 42.29 | 22k |
| Total | 683 | 21 | 282.60 | 70k |

Table 13: Existing assets grouped by license.

| License | Assets |
|---------|--------|
| CC-BY-SA-4.0 | TQA, MapQA, MathVista, AI2D |
| Apache-2.0 | DriveLM, MathV360k, CV-Bench, CoIN |
| MIT | Monkey, OCRbench, MAVIS |
| CC-BY-4.0 | CLEVR, ScreenQA, Screen2Words, MMTBench |

**Statistics of StockQA dataset.** StockQA is a new VQA dataset related to multimodal stock analysis. It includes 70k question-answer pairs. of which 60k is the training set and 10k is the test set. For the

training data, there are 40k MCQ and 20k Y/N QA pairs. For the test data, there are 8k MCQ and 2k QA pairs. Each choice is equally distributed after our cleaning process. Figs. 13 and 14 shows the word cloud and examples of StockQA dataset. Tab. 12 and fig. 12 shows the detailed statistics of StockQA dataset.

**Dataset License.** Our dataset follows the CC-BY license. This license allows reusers to distribute, remix, adapt, and build upon the material in any medium or format, so long as attribution is given to the creator. The license allows for commercial use. For other assets we used, we list the licenses below in Tab. 13.

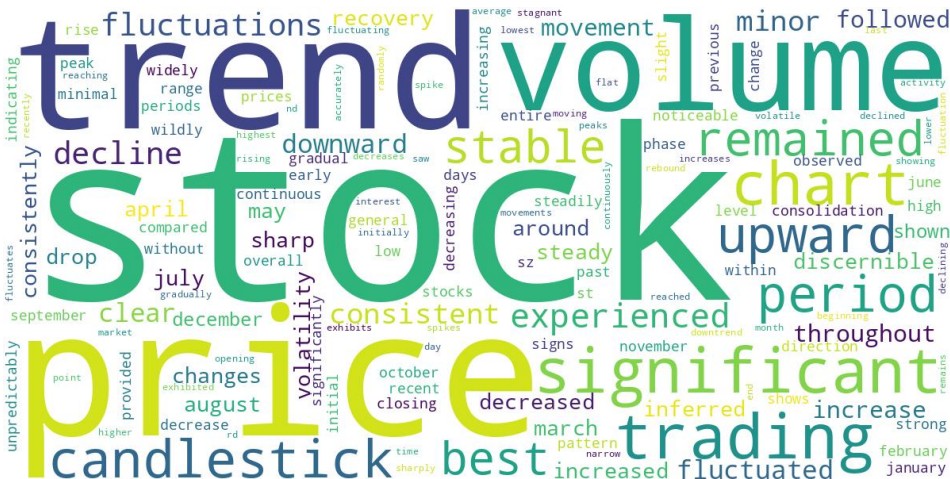

Figure 14: Word cloud of StockQA dataset.

## C  DETAILED CONTINUAL LEARNING RESULTS

In this section, we show the detailed inference results of all the methods (LoRA (Hu et al., 2021), LoRA* (Hu et al., 2021), O-LoRA (Wang et al., 2023a), O-LoRA* (Wang et al., 2023a), MoELoRA (Chen et al., 2024a), MoELoRA* (Chen et al., 2024a), CL-MoE (Huai et al., 2025), CL-MoE* (Huai et al., 2025), HiDe (Guo et al., 2025a), HiDe* (Guo et al., 2025a), SEFE (Chen et al., 2025), SEFE* (Chen et al., 2025), DISCO (Guo et al., 2025b), DISCO* (Guo et al., 2025b) and MR-LoRA) during each continual learning stage, where * denotes the original method with replay data.

### C.1  BASELINE RESULTS IN DOMAIN CONTINUAL LEARNING

Table 14: Result matrices of InternVL-based baselines in domain continual learning. * denotes the original method with replay data.

| LoRA-FT | RS | Med | AD | Sci | Fin |
|---|---|---|---|---|---|
| RS | 81.29 | | | | |
| Med | 75.71 | 65.92 | | | |
| AD | 69.38 | 56.87 | 53.56 | | |
| Sci | 71.12 | 53.75 | 46.83 | 53.48 | |
| Fin | 69.93 | 52.17 | 33.04 | 42.67 | 91.07 |

| LoRA-FT* | RS | Med | AD | Sci | Fin |
|---|---|---|---|---|---|
| RS | 81.68 | | | | |
| Med | 77.45 | 66.69 | | | |
| AD | 77.24 | 61.32 | 53.81 | | |
| Sci | 77.89 | 55.43 | 49.13 | 53.53 | |
| Fin | 77.06 | 47.55 | 42.67 | 43.31 | 91.44 |

| MoELoRA | RS | Med | AD | Sci | Fin |
|---|---|---|---|---|---|
| RS | 81.22 | | | | |
| Med | 77.56 | 66.00 | | | |
| AD | 74.56 | 58.74 | 53.62 | | |
| Sci | 72.62 | 54.77 | 47.65 | 52.75 | |
| Fin | 69.90 | 52.08 | 33.17 | 42.19 | 90.58 |

| MoELoRA* | RS | Med | AD | Sci | Fin |
|---|---|---|---|---|---|
| RS | 80.75 | | | | |
| Med | 78.10 | 64.77 | | | |
| AD | 73.24 | 59.54 | 52.90 | | |
| Sci | 76.82 | 53.64 | 42.11 | 51.24 | |
| Fin | 76.74 | 52.65 | 38.81 | 42.15 | 89.84 |

| HiDe | RS | Med | AD | Sci | Fin |
|---|---|---|---|---|---|
| RS | 81.24 | | | | |
| Med | 79.59 | 64.71 | | | |
| AD | 78.85 | 58.37 | 41.75 | | |
| Sci | 78.33 | 58.51 | 39.94 | 49.99 | |
| Fin | 75.40 | 57.66 | 36.73 | 41.48 | 88.59 |

| HiDe* | RS | Med | AD | Sci | Fin |
|---|---|---|---|---|---|
| RS | 73.92 | | | | |
| Med | 71.44 | 64.22 | | | |
| AD | 65.74 | 49.83 | 42.92 | | |
| Sci | 70.64 | 54.79 | 40.47 | 50.75 | |
| Fin | 53.17 | 52.61 | 40.85 | 47.04 | 89.17 |

| DISCO | RS | Med | AD | Sci | Fin |
|---|---|---|---|---|---|
| RS | 81.42 | | | | |
| Med | 79.13 | 63.80 | | | |
| AD | 78.62 | 60.79 | 53.98 | | |
| Sci | 77.40 | 52.21 | 53.74 | 54.18 | |
| Fin | 75.12 | 50.69 | 52.41 | 50.67 | 90.86 |

| DISCO* | RS | Med | AD | Sci | Fin |
|---|---|---|---|---|---|
| RS | 81.49 | | | | |
| Med | 80.14 | 63.05 | | | |
| AD | 78.87 | 57.42 | 53.77 | | |
| Sci | 78.67 | 52.80 | 53.56 | 53.52 | |
| Fin | 77.90 | 47.50 | 49.13 | 49.37 | 90.92 |

Table 15: Result matrices of LLaVA-based baselines in domain continual learning. * denotes the original method with replay data.

| LoRA-FT | RS | Med | AD | Sci | Fin |
|---|---|---|---|---|---|
| RS | 78.32 | | | | |
| Med | 74.68 | 57.53 | | | |
| AD | 68.93 | 47.19 | 52.15 | | |
| Sci | 75.12 | 45.56 | 38.46 | 49.44 | |
| Fin | 69.65 | 41.59 | 25.43 | 40.88 | 87.45 |

| LoRA-FT* | RS | Med | AD | Sci | Fin |
|---|---|---|---|---|---|
| RS | 79.33 | | | | |
| Med | 76.45 | 57.58 | | | |
| AD | 74.54 | 54.26 | 52.96 | | |
| Sci | 77.00 | 50.31 | 45.13 | 51.88 | |
| Fin | 76.54 | 50.27 | 43.01 | 43.32 | 89.85 |

| O-LoRA | RS | Med | AD | Sci | Fin |
|---|---|---|---|---|---|
| RS | 79.25 | | | | |
| Med | 74.05 | 56.52 | | | |
| AD | 76.06 | 43.71 | 52.32 | | |
| Sci | 76.60 | 44.87 | 40.57 | 50.58 | |
| Fin | 74.64 | 44.42 | 30.02 | 41.47 | 87.15 |

| O-LoRA* | RS | Med | AD | Sci | Fin |
|---|---|---|---|---|---|
| RS | 79.17 | | | | |
| Med | 78.21 | 56.65 | | | |
| AD | 77.52 | 38.60 | 37.81 | | |
| Sci | 77.61 | 44.22 | 35.40 | 45.59 | |
| Fin | 76.94 | 41.17 | 34.18 | 39.61 | 83.22 |

| MoELoRA | RS | Med | AD | Sci | Fin |
|---|---|---|---|---|---|
| RS | 79.09 | | | | |
| Med | 74.78 | 58.73 | | | |
| AD | 77.69 | 43.72 | 51.47 | | |
| Sci | 76.87 | 43.79 | 32.81 | 48.67 | |
| Fin | 77.54 | 41.85 | 27.62 | 40.13 | 86.75 |

| MoELoRA* | RS | Med | AD | Sci | Fin |
|---|---|---|---|---|---|
| RS | 79.66 | | | | |
| Med | 78.44 | 60.50 | | | |
| AD | 78.54 | 49.86 | 52.54 | | |
| Sci | 78.00 | 50.53 | 43.32 | 49.30 | |
| Fin | 77.63 | 49.54 | 39.08 | 41.04 | 89.21 |

| CL-MoE | RS | Med | AD | Sci | Fin |
|---|---|---|---|---|---|
| RS | 79.08 | | | | |
| Med | 73.48 | 60.56 | | | |
| AD | 72.61 | 44.42 | 51.62 | | |
| Sci | 71.02 | 48.04 | 37.70 | 50.28 | |
| Fin | 71.34 | 46.84 | 26.33 | 41.17 | 88.74 |

| CL-MoE* | RS | Med | AD | Sci | Fin |
|---|---|---|---|---|---|
| RS | 79.40 | | | | |
| Med | 76.32 | 61.10 | | | |
| AD | 72.01 | 54.49 | 52.56 | | |
| Sci | 76.64 | 53.89 | 43.83 | 49.98 | |
| Fin | 76.58 | 52.31 | 39.65 | 45.64 | 90.21 |

| HiDe | RS | Med | AD | Sci | Fin |
|---|---|---|---|---|---|
| RS | 78.14 | | | | |
| Med | 74.26 | 58.05 | | | |
| AD | 74.90 | 42.94 | 39.65 | | |
| Sci | 75.43 | 44.91 | 38.33 | 46.44 | |
| Fin | 74.31 | 48.95 | 33.21 | 38.54 | 81.55 |

| HiDe* | RS | Med | AD | Sci | Fin |
|---|---|---|---|---|---|
| RS | 79.21 | | | | |
| Med | 77.79 | 60.88 | | | |
| AD | 77.64 | 48.19 | 38.12 | | |
| Sci | 77.51 | 48.84 | 35.76 | 46.71 | |
| Fin | 74.80 | 42.29 | 34.03 | 38.01 | 79.22 |

| SEFE | RS | Med | AD | Sci | Fin |
|---|---|---|---|---|---|
| RS | 78.27 | | | | |
| Med | 76.32 | 58.42 | | | |
| AD | 77.22 | 49.13 | 52.49 | | |
| Sci | 77.83 | 47.70 | 43.01 | 49.04 | |
| Fin | 77.26 | 50.37 | 37.21 | 40.87 | 86.82 |

| SEFE* | RS | Med | AD | Sci | Fin |
|---|---|---|---|---|---|
| RS | 79.21 | | | | |
| Med | 78.39 | 60.93 | | | |
| AD | 79.00 | 57.68 | 53.11 | | |
| Sci | 78.76 | 51.39 | 47.99 | 51.87 | |
| Fin | 78.43 | 52.85 | 46.21 | 47.76 | 89.33 |

| DISCO | RS | Med | AD | Sci | Fin |
|---|---|---|---|---|---|
| RS | 78.57 | | | | |
| Med | 75.80 | 52.36 | | | |
| AD | 76.37 | 49.78 | 53.04 | | |
| Sci | 76.11 | 45.76 | 49.26 | 49.23 | |
| Fin | 76.03 | 45.20 | 43.79 | 42.33 | 88.95 |

| DISCO* | RS | Med | AD | Sci | Fin |
|---|---|---|---|---|---|
| RS | 79.20 | | | | |
| Med | 77.96 | 52.44 | | | |
| AD | 78.05 | 49.85 | 53.03 | | |
| Sci | 77.26 | 46.32 | 53.08 | 51.99 | |
| Fin | 77.78 | 46.25 | 50.45 | 49.51 | 89.71 |

## C.2 BASELINE RESULTS IN ABILITY CONTINUAL LEARNING

Table 16: Result matrices of LLaVA-based baselines in ability continual learning. $*$ denotes the original method with replay data.

| LoRA-FT | OCR | Math | VP | APP |
|---|---|---|---|---|
| OCR | 33.30 | | | |
| Math | 32.60 | 34.20 | | |
| VP | 31.70 | 32.80 | 65.10 | |
| APP | 23.60 | 33.70 | 55.84 | 32.50 |

| LoRA-FT$^*$ | OCR | Math | VP | APP |
|---|---|---|---|---|
| OCR | 32.60 | | | |
| Math | 33.60 | 33.80 | | |
| VP | 31.10 | 33.50 | 66.12 | |
| APP | 21.80 | 32.70 | 58.38 | 28.75 |

| O-LoRA | OCR | Math | VP | APP |
|---|---|---|---|---|
| OCR | 32.90 | | | |
| Math | 29.80 | 33.60 | | |
| VP | 27.40 | 33.70 | 58.63 | |
| APP | 29.60 | 32.90 | 52.41 | 33.75 |

| O-LoRA$^*$ | OCR | Math | VP | APP |
|---|---|---|---|---|
| OCR | 34.00 | | | |
| Math | 28.40 | 36.80 | | |
| VP | 28.90 | 33.90 | 61.55 | |
| APP | 29.60 | 31.30 | 60.79 | 27.50 |

| MoELoRA | OCR | Math | VP | APP |
|---|---|---|---|---|
| OCR | 32.70 | | | |
| Math | 32.50 | 33.30 | | |
| VP | 30.80 | 33.00 | 64.59 | |
| APP | 26.70 | 32.80 | 56.85 | 27.22 |

| MoELoRA$^*$ | OCR | Math | VP | APP |
|---|---|---|---|---|
| OCR | 32.70 | | | |
| Math | 29.40 | 33.10 | | |
| VP | 32.60 | 32.50 | 65.61 | |
| APP | 19.80 | 32.20 | 54.19 | 30.00 |

| CL-MoE | OCR | Math | VP | APP |
|---|---|---|---|---|
| OCR | 33.00 | | | |
| Math | 32.30 | 33.60 | | |
| VP | 30.20 | 32.50 | 64.72 | |
| APP | 19.90 | 32.70 | 53.43 | 30.69 |

| CL-MoE$^*$ | OCR | Math | VP | APP |
|---|---|---|---|---|
| OCR | 33.20 | | | |
| Math | 34.30 | 36.70 | | |
| VP | 32.00 | 33.20 | 64.97 | |
| APP | 25.40 | 31.80 | 60.91 | 30.00 |

| HiDe | OCR | Math | VP | APP |
|---|---|---|---|---|
| OCR | 33.40 | | | |
| Math | 30.90 | 32.80 | | |
| VP | 30.40 | 33.30 | 56.98 | |
| APP | 24.60 | 32.10 | 46.32 | 28.75 |

| HiDe$^*$ | OCR | Math | VP | APP |
|---|---|---|---|---|
| OCR | 34.10 | | | |
| Math | 32.60 | 35.70 | | |
| VP | 30.70 | 32.60 | 53.81 | |
| APP | 24.60 | 28.40 | 30.71 | 23.75 |

| SEFE | OCR | Math | VP | APP |
|---|---|---|---|---|
| OCR | 33.00 | | | |
| Math | 32.20 | 32.60 | | |
| VP | 31.80 | 33.30 | 64.59 | |
| APP | 26.00 | 33.40 | 57.74 | 33.75 |

| SEFE$^*$ | OCR | Math | VP | APP |
|---|---|---|---|---|
| OCR | 33.60 | | | |
| Math | 33.80 | 37.50 | | |
| VP | 32.80 | 36.10 | 66.50 | |
| APP | 25.60 | 34.80 | 57.61 | 31.39 |

| DISCO | OCR | Math | VP | APP |
|---|---|---|---|---|
| OCR | 32.90 | | | |
| Math | 31.80 | 33.40 | | |
| VP | 31.00 | 34.50 | 59.64 | |
| APP | 32.90 | 33.10 | 60.15 | 30.14 |

| DISCO$^*$ | OCR | Math | VP | APP |
|---|---|---|---|---|
| OCR | 33.40 | | | |
| Math | 32.10 | 36.60 | | |
| VP | 32.20 | 37.00 | 63.07 | |
| APP | 34.20 | 35.00 | 61.55 | 27.50 |

Table 17: Result matrices of InternVL-based baselines in ability continual learning. * denotes the original method with replay data.

| LoRA-FT | OCR | Math | VP | APP |
|---|---|---|---|---|
| OCR | 32.20 | | | |
| Math | 33.10 | 33.30 | | |
| VP | 31.80 | 32.30 | 68.02 | |
| APP | 21.40 | 32.80 | 60.28 | 29.86 |

| LoRA-FT* | OCR | Math | VP | APP |
|---|---|---|---|---|
| OCR | 31.60 | | | |
| Math | 35.30 | 35.40 | | |
| VP | 32.60 | 31.10 | 68.27 | |
| APP | 26.30 | 34.20 | 62.56 | 31.25 |

| O-LoRA | OCR | Math | VP | APP |
|---|---|---|---|---|
| OCR | 32.70 | | | |
| Math | 31.10 | 34.20 | | |
| VP | 30.20 | 33.00 | 63.20 | |
| APP | 25.50 | 32.30 | 64.59 | 24.44 |

| O-LoRA* | OCR | Math | VP | APP |
|---|---|---|---|---|
| OCR | 34.00 | | | |
| Math | 30.90 | 34.40 | | |
| VP | 31.00 | 33.20 | 65.86 | |
| APP | 21.70 | 31.10 | 59.77 | 31.25 |

| MoELoRA | OCR | Math | VP | APP |
|---|---|---|---|---|
| OCR | 32.20 | | | |
| Math | 29.90 | 33.30 | | |
| VP | 29.20 | 32.80 | 67.64 | |
| APP | 17.20 | 32.70 | 55.33 | 32.50 |

| MoELoRA* | OCR | Math | VP | APP |
|---|---|---|---|---|
| OCR | 32.90 | | | |
| Math | 31.50 | 36.50 | | |
| VP | 30.90 | 32.30 | 65.74 | |
| APP | 13.90 | 29.70 | 54.95 | 32.50 |

| HiDe | OCR | Math | VP | APP |
|---|---|---|---|---|
| OCR | 33.40 | | | |
| Math | 26.30 | 33.60 | | |
| VP | 30.10 | 33.00 | 61.80 | |
| APP | 17.70 | 33.00 | 41.12 | 20.28 |

| HiDe* | OCR | Math | VP | APP |
|---|---|---|---|---|
| OCR | 33.40 | | | |
| Math | 28.10 | 34.70 | | |
| VP | 31.10 | 32.20 | 55.33 | |
| APP | 25.30 | 29.20 | 42.13 | 20.28 |

| DISCO | OCR | Math | VP | APP |
|---|---|---|---|---|
| OCR | 31.90 | | | |
| Math | 31.70 | 34.00 | | |
| VP | 32.10 | 33.50 | 63.45 | |
| APP | 30.60 | 33.10 | 65.36 | 27.50 |

| DISCO* | OCR | Math | VP | APP |
|---|---|---|---|---|
| OCR | 34.70 | | | |
| Math | 31.50 | 34.70 | | |
| VP | 31.50 | 34.60 | 62.31 | |
| APP | 32.30 | 32.30 | 64.97 | 30.14 |

## C.3 DETAILED RESULTS OF MR-LoRA

Table 18: Result matrices of MR-LoRA in domain continual learning. LLaVA denotes LLaVA-based MR-LoRA, and InternVL denotes InternVL-based MR-LoRA.

| LLaVA | RS | Med | AD | Sci | Fin |
|---|---|---|---|---|---|
| RS | 81.06 | | | | |
| Med | 81.06 | 65.73 | | | |
| AD | 81.06 | 65.71 | 54.17 | | |
| Sci | 81.06 | 65.68 | 54.17 | 56.11 | |
| Fin | 80.87 | 65.32 | 54.12 | 56.71 | 91.12 |

| InternVL | RS | Med | AD | Sci | Fin |
|---|---|---|---|---|---|
| RS | 81.49 | | | | |
| Med | 81.49 | 66.40 | | | |
| AD | 81.49 | 66.42 | 54.56 | | |
| Sci | 81.47 | 65.81 | 54.56 | 54.05 | |
| Fin | 81.48 | 65.80 | 54.56 | 54.40 | 91.07 |

Table 19: Result matrices of MR-LoRA in ability continual learning. LLaVA denotes LLaVA-based MR-LoRA, and InternVL denotes InternVL-based MR-LoRA.

| LLaVA | OCR | Math | VP | APP |
|---|---|---|---|---|
| OCR | 33.60 | | | |
| Math | 33.50 | 36.50 | | |
| VP | 33.50 | 36.40 | 64.97 | |
| APP | 33.70 | 36.20 | 65.10 | 32.50 |

| InternVL | OCR | Math | VP | APP |
|---|---|---|---|---|
| OCR | 32.20 | | | |
| Math | 33.80 | 36.40 | | |
| VP | 33.30 | 35.60 | 67.89 | |
| APP | 33.00 | 35.70 | 67.51 | 33.75 |

Table 20: Comparison of different methods on the UCIT benchmark. MR-LoRA still remains competitive performance.

| Method | Venue | ImgNet-R | ArxivQA | VizWiz | IconQA | CLEVR | Flickr30k | MFT (↑) | MFN (↑) | MAA (↑) | BWT (↑) |
|---|---|---|---|---|---|---|---|---|---|---|---|
| Zero-shot | - | 16.27 | 53.73 | 38.39 | 19.20 | 20.63 | 41.88 | - | 31.68 | - | - |
| Oracle | - | 91.37 | 94.27 | 60.19 | 78.13 | 78.77 | 57.79 | - | 76.75 | - | - |
| LoRA-FT | ICLR-22 | 58.03 | 77.63 | 44.39 | 67.40 | 61.77 | **58.22** | **76.89** | 61.24 | 76.55 | -18.78 |
| O-LoRA | EMNLP-23 | 77.50 | 78.07 | 44.50 | 63.13 | 64.73 | 58.16 | 76.01 | 64.35 | 78.02 | -13.99 |
| MoELoRA | NeurIPS-24 | 70.07 | 77.70 | 44.69 | 50.03 | 54.03 | 57.34 | 71.17 | 58.98 | 75.08 | -14.63 |
| ModalPrompt | EMNLP-25 | 51.07 | 87.27 | 48.11 | 39.23 | 46.57 | 42.93 | 52.65 | 52.53 | 57.67 | **-0.15** |
| CL-MoE | CVPR-25 | 66.33 | 77.00 | 44.78 | 51.87 | 53.53 | 57.42 | 71.46 | 58.49 | 74.19 | -15.56 |
| HiDE | ACL-25 | 84.03 | 90.73 | 44.43 | 58.93 | 41.37 | 54.25 | 69.96 | 62.29 | 77.32 | -9.20 |
| SEFE | ICML-25 | 80.83 | 78.00 | 47.01 | 69.63 | 65.83 | 57.92 | 75.98 | 66.54 | 78.76 | -11.33 |
| DISCO | ICCV-25 | 87.43 | **93.07** | 46.96 | 68.13 | 65.70 | 56.69 | 75.87 | 69.66 | 81.60 | -7.45 |
| MR-LoRA | Ours | **90.20** | 91.20 | **56.34** | **78.00** | **76.13** | 56.34 | 76.19 | **74.70** | **82.84** | -1.79 |

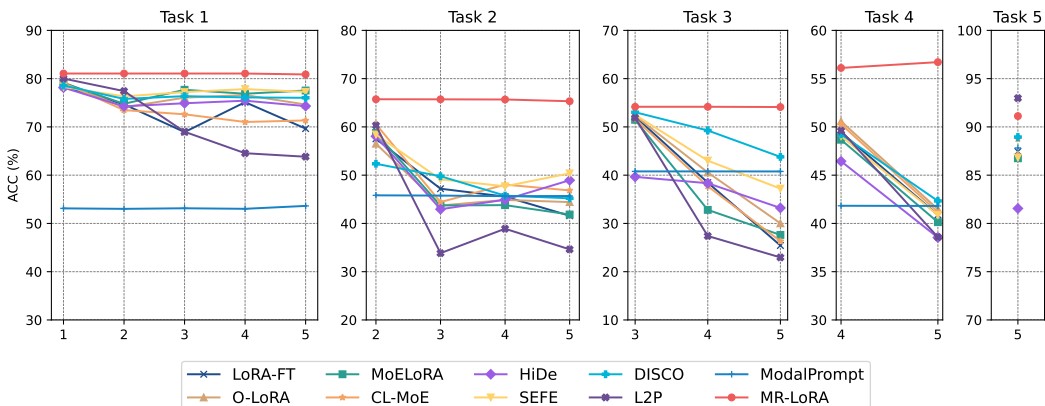

Figure 15: Visualization of LLaVA DCL without replay. Red line is MR-LoRA. Higher is better.

## C.4 RESULTS ON UCIT BENCHMARK

To further demonstrate that MR-LoRA's competitive performance, we conduct MR-LoRA on UCIT (Guo et al., 2025a) benchmark. The results are shown in Tab. 20. The results on UCIT demonstrate that MR-LoRA remains highly competitive and outperforms existing state-of-the-art methods, even on datasets where task heterogeneity is less pronounced than in MLLM-CL, *i.e.*, dataset incremental learning.

## C.5 VISUALIZATION OF RESULTS

Figs. 15 to 22 provide a clear graphical overview of our model's performance across the different domains and abilities compared to baselines. Each subfigure dedicated to tracking the performance of one specific task (*e.g.*, Subfigure 1 (heading Task 1) for RS in DCL, Subfigure 2 (heading Task 2) for Medical in DCL, *etc.*) throughout the entire continual learning process.

## C.6 FURTHER DISCUSSION

**Order-agnostic.** MR-LoRA employs a parameter isolation strategy that ensures the model remains order-agnostic. Specifically: (1) Independent expert initialization: For each distinct task (*e.g.*, Remote Sensing or Medical), MR-LoRA instantiates a dedicated LoRA module. Crucially, this module undergoes independent initialization rather than inheriting parameters optimized for preceding tasks. (2) Decoupled optimization: Since the parameters for Task B are not derived from those of Task

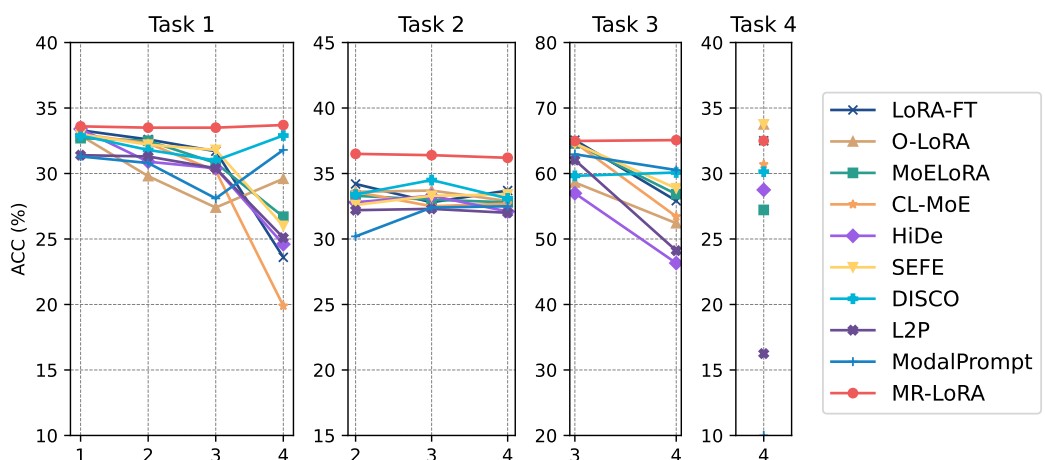

Figure 16: Visualization of LLaVA ACL without replay. Red line is MR-LoRA. Higher is better.

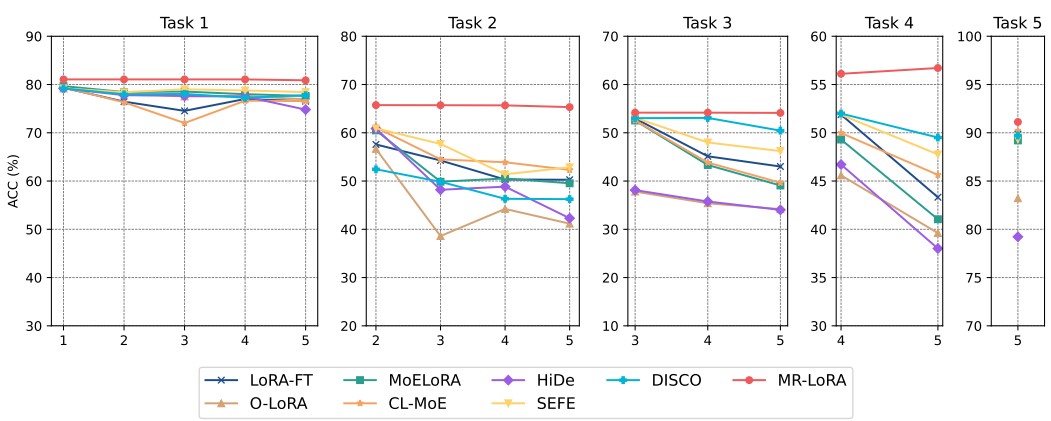

Figure 17: Visualization of LLaVA DCL with replay. Red line is MR-LoRA. Higher is better.

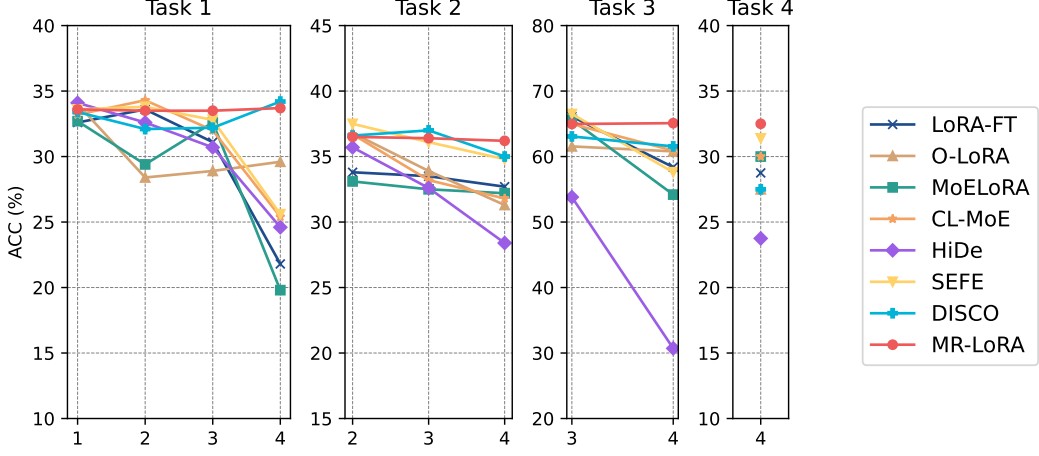

Figure 18: Visualization of LLaVA ACL with replay. Red line is MR-LoRA. Higher is better.

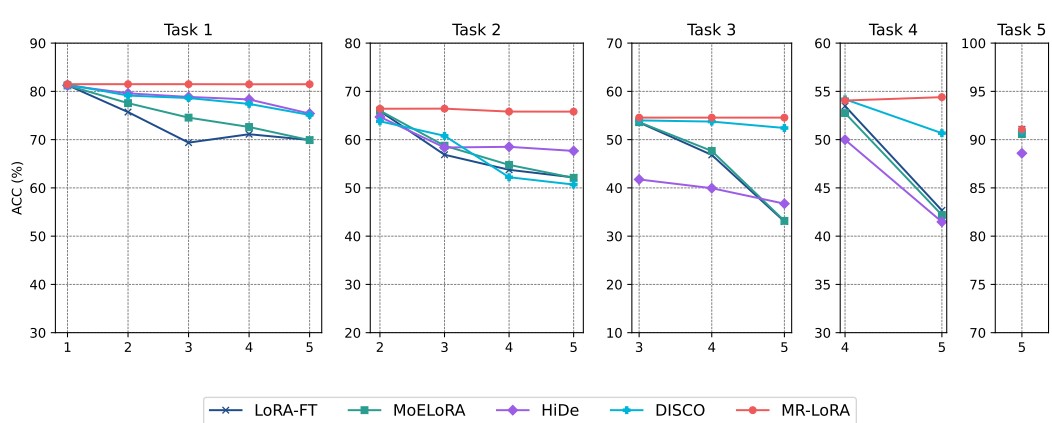

Figure 19: Visualization of InternVL DCL without replay. Red line is MR-LoRA. Higher is better.

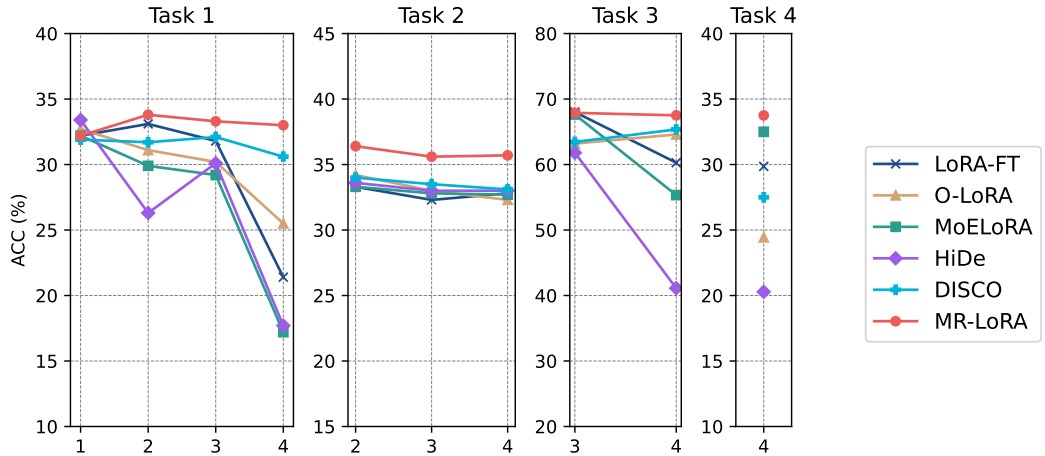

Figure 20: Visualization of InternVL ACL without replay. Red line is MR-LoRA. Higher is better.

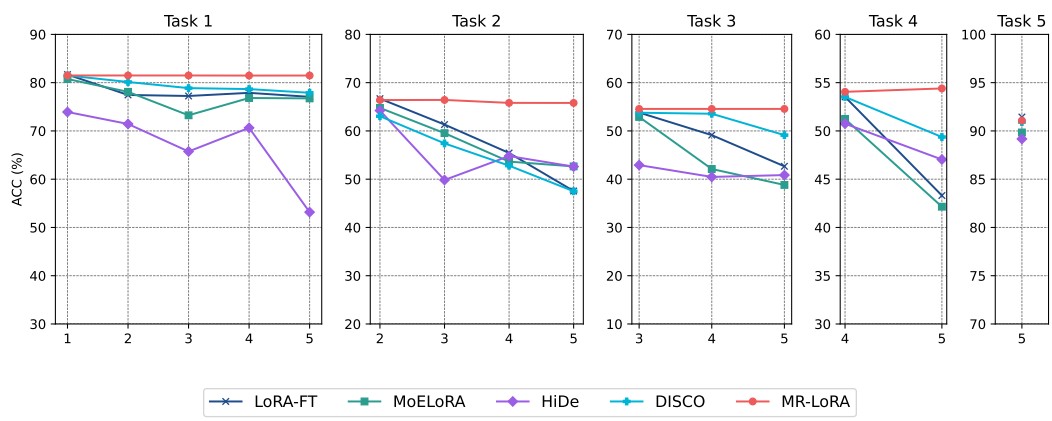

Figure 21: Visualization of InternVL DCL with replay. Red line is MR-LoRA. Higher is better.

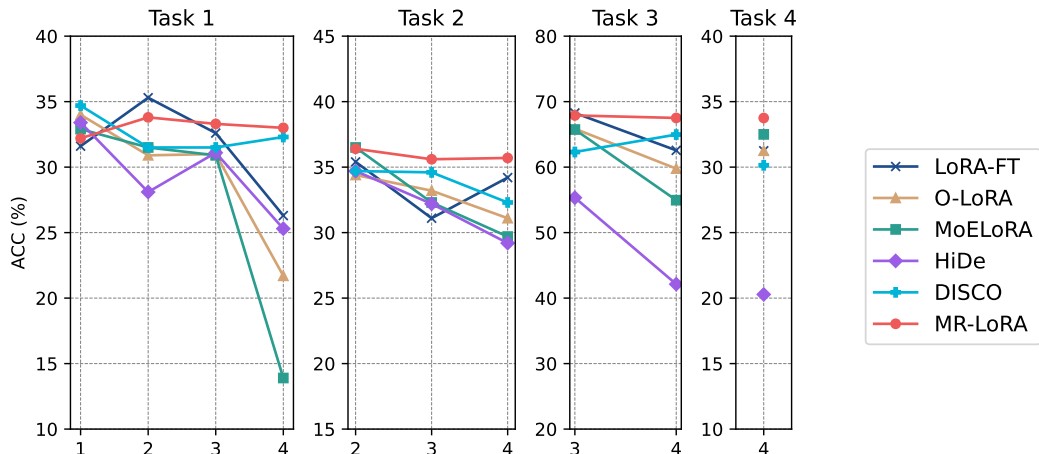

Figure 22: Visualization of InternVL ACL with replay. Red line is MR-LoRA. Higher is better.

A, the optimization trajectory for Task B remains invariant to the learning sequence of Task A. While the router undergoes sequential updates, it leverages a few-shot strategy to map inputs to the independently trained experts. Because these experts preserve their plasticity and specificity irrespective of the curriculum order, the system circumvents the "plasticity-stability" trade-off that typically induces order sensitivity in shared-parameter models. When tasks exhibit high similarity or overlap, a primary concern is that the router may struggle to make consistent decisions. In the case of two identical or highly similar tasks (*e.g.*, Task A and Task F are the same), we would train two separate experts, expert A and expert F. Because they are trained on the same data distribution, these two experts will become functionally equivalent. When a new input from this task distribution is presented during inference, the router's objective is to pick an expert that maximizes performance. Since both expert A and expert F are equally capable of handling the input, the router can select either one and still achieve a high score. Therefore, what might seem like a "struggle" or inconsistency in choice is actually a reflection of the task redundancy. The router correctly identifies that multiple experts are suitable, and its final choice does not degrade performance. While this leads to a degree of parameter redundancy (storing two nearly identical experts), it crucially avoids the primary failure mode of other methods: catastrophic forgetting or negative transfer. Our method prioritizes knowledge preservation, and the architectural isolation of experts ensures that learning a redundant task does not corrupt other distinct experts. We will consider the promising strategies as our future work (*Appendix* D.2) to avoid this parameter redundancy.

**MR-LoRA can work well when tasks are highly similar or overlapping.** To empirically validate this, we have conducted a new experiment and extended the Domain Continual Learning (DCL) setting by adding a sixth task that is an exact repetition of the first task, Remote Sensing (RS). Now, the new task sequence is: **RS → Medical → Autonomous Driving → Science → Finance → RS (repeat)**. The experiments are shown in Tab. 21, we can find that MR-LoRA can still work very well under this scenario.

Table 21: MR-LoRA can work well when tasks are highly similar or overlapping (in DCL, LLaVA).

| Method | $RS_1$ | Med | AD | Sci | Fin | $RS_2$ |
|---|---|---|---|---|---|---|
| MR-LoRA (5 tasks) | 80.87 | 65.32 | 54.12 | 56.71 | 91.12 | - |
| MR-LoRA (6 tasks) | 81.02 | 65.83 | 54.17 | 55.94 | 91.11 | 81.02 |

# D  LIMITATIONS AND BROADER IMPACTS

## D.1  LIMITATIONS

Although our study makes valuable contributions, we acknowledge the following limitations: (1) Model size and training limitations: This research focuses exclusively on MLLMs with 7 billion parameters. Owing to computational constraints, we did not explore larger models. (2) Potential inaccuracies in the StockQA dataset: Our StockQA dataset is generated by Qwen2.5-VL (Bai et al., 2025), and the model may inadvertently produce inaccurate or misleading data. Moreover, biases inherent in the training data could manifest in the generated dataset, influencing the outcomes and interpretations of subsequent analyses. (3) Linear parameter growth: Although MR-LoRA is parameter-efficient (only adding around 1% parameter for each new task), its linear parameter growth is a critical consideration for very long-term continual learning. We will discuss potential mitigation strategies in *Appendix* D.2.  We hope to address these limitations in our future work to build a practical and lifelong-evolving MLLM.

## D.2  FUTURE WORK

While our MR-LoRA framework effectively mitigates catastrophic forgetting, its linear parameter growth with each new task presents a scalability challenge for truly long-term, open-ended learning scenarios. We identify two key avenues for future work: enhancing parameter efficiency and advancing towards open-world learning. To address the scalability concern in scenarios involving a vast number of tasks, we propose several strategies to optimize parameter efficiency:

- **Expert Merging:** Periodically fusing similar LoRA modules based on the router's similarity scores to reduce the total count of stored experts.
- **Hierarchical/Shared Adapters:** Implementing a "library" of shared LoRA bases where new tasks only learn a lightweight delta, rather than a full independent adapter.
- **Dynamic Pruning:** Removing or compressing rarely accessed experts over time.

A more fundamental and ambitious extension is to move from our current continual learning setting towards Open-World Continual Learning. Our MLLM-CL benchmark operates under the assumption that new tasks are explicitly defined and belong to a known sequence. A truly autonomous MLLM, however, must operate in an open world where it encounters data from entirely novel, unforeseen domains or requires abilities it was never trained on. This paradigm introduces several profound challenges:

- **Novelty Detection:** The MLLM-based router must be enhanced to not only select the best-suited known expert but also to recognize when an input does not fit any existing expertise. This requires endowing the router with a robust out-of-distribution detection or "rejection" capability, allowing it to identify inputs from unknown tasks.
- **Automated Expert Creation and Learning:** Upon detecting a consistent stream of novel inputs, the system should be able to automatically instantiate a new expert module and initiate a learning process. This may involve unsupervised clustering or routing confidence of the new data to form a nascent task representation, followed by self-supervised or few-shot learning to build the new expert's capability without human intervention.

Successfully addressing these open-world challenges would represent a significant leap towards creating genuinely lifelong-learning MLLMs that can autonomously adapt, expand their knowledge, and grow their skill set in unconstrained, dynamic environments.

## D.3  BROADER IMPACTS

Positively, such work advances the ability of AI systems to learn adaptively from ongoing streams of diverse data, enabling applications in education, assistive technologies, and personalized healthcare.

These systems could provide more context-aware and accessible tools that evolve over time to better support users' needs. Moreover, robust continual learning reduces the need for retraining from scratch, leading to more energy-efficient and sustainable AI development. However, there are potential negative impacts. Without careful design, continual learning systems may inadvertently retain or amplify biases from evolving data streams, leading to fairness concerns. The dynamic nature of these models also complicates auditing and accountability, as their behavior changes over time. Additionally, if misused, adaptive models could enhance surveillance or manipulation by continuously tailoring outputs to influence user behavior. To mitigate these risks, transparency, rigorous evaluation, and ethical safeguards must be integrated into both benchmark design and method development.

## E    INFERENCE OPTIMIZATION WITH CACHING

A key advantage of our method is its computational efficiency during inference. While our approach involves two distinct phases, we introduce a caching strategy that collapses the computational overhead. The most intensive operation—the forward pass through the backbone network (*i.e.*, the visual encoder and LLM) is performed only once. We cache the resulting hidden states from each layer (specifically, the KV cache) after this single pass. Subsequently, our two lightweight modules, the router and the expert LoRA, operate sequentially on these cached states, obviating the need for a second full forward pass. This optimization reduces the computational cost from that of two full inferences to only marginally more than a single one, achieving a practical deployment cost comparable to standard single-pass methods, such as LoRA-FT (Hu et al., 2021).

## F    USE OF LLM

In the preparation of this manuscript, we utilized a Large Language Model (LLM) in a capacity analogous to a conventional grammar-checking tool. Its application was strictly confined to copy-editing tasks, such as correcting spelling, improving grammar, and enhancing the clarity of author-generated text. No part of the research ideation, methodology, data analysis, or generation of substantive content was performed by the LLM.

# G VISUALIZATION

## G.1 ILLUSTRATION OF MLLM-CL BENCHMARK

In this section, we show more examples of our MLLM-CL benchmark in domain continual learning and ability continual learning.

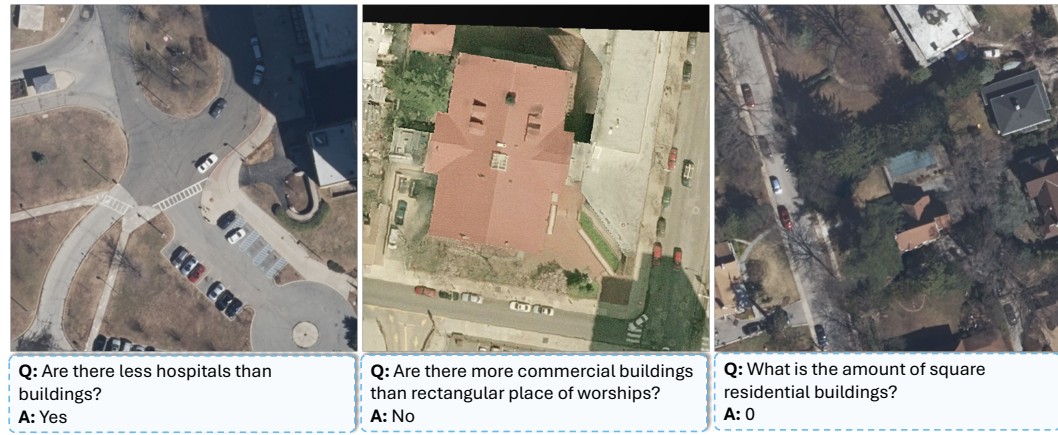

Figure 23: Examples of remote sensing task in domain continual learning.

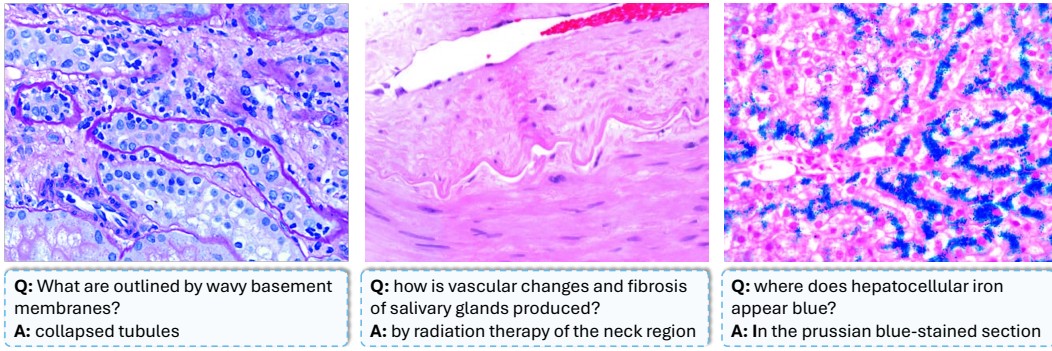

Figure 24: Examples of medical task in domain continual learning.

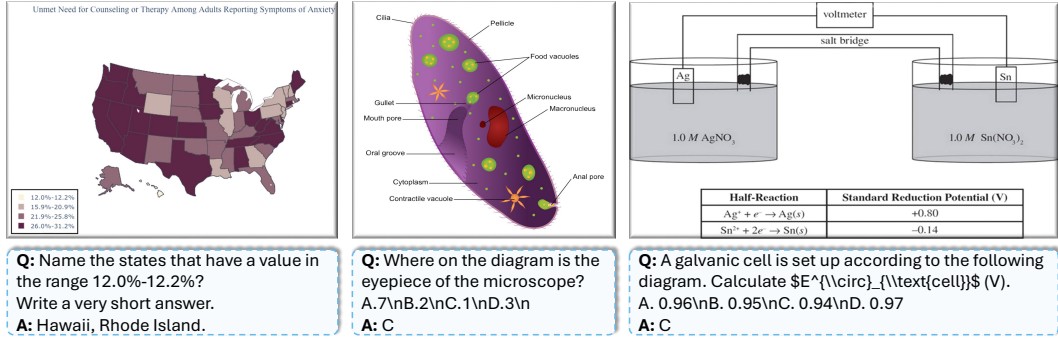

Figure 25: Examples of science task in domain continual learning.

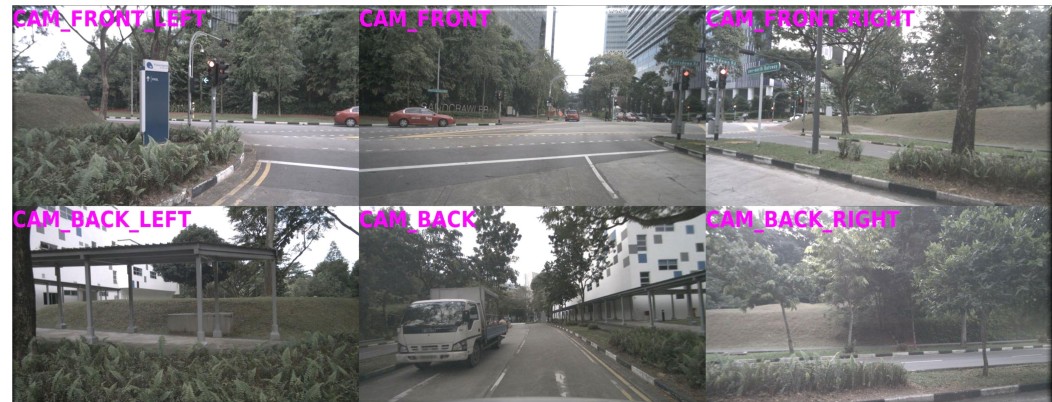

**Q:** What is the future state of <c1,CAM_FRONT,[539, 273]>? Objects are encoded using <c,CAM,[cx,cy]>, where c is the identifier, CAM indicates the camera where the object's center point is situated, and x, y represent the horizontal and vertical coordinates of the center point of the 2D bounding box.
**A:** Keep going straight.

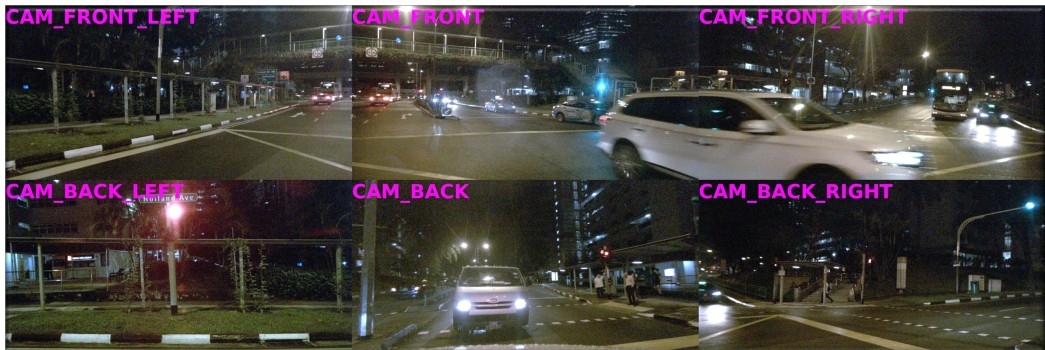

**Q:** Is there any traffic element in the front view? Objects are encoded using <c,CAM,[cx,cy]>, where c is the identifier, CAM indicates the camera where the object's center point is situated, and x, y represent the horizontal and vertical coordinates of the center point of the 2D bounding box.
**A:** Yes, there are some traffic elements in the front view.

Figure 26: Examples of autonomous driving task in domain continual learning.

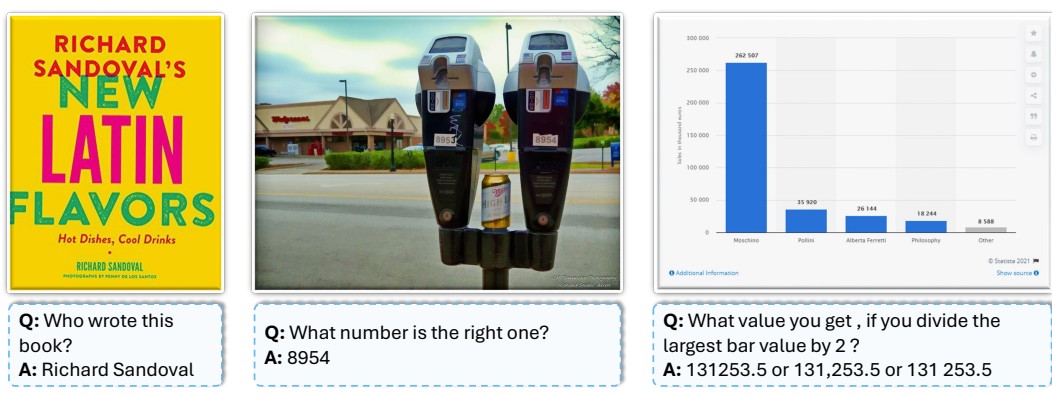

**Q:** Who wrote this book?
**A:** Richard Sandoval

**Q:** What number is the right one?
**A:** 8954

**Q:** What value you get , if you divide the largest bar value by 2 ?
**A:** 131253.5 or 131,253.5 or 131 253.5

Figure 27: Examples of OCR task in ability continual learning.

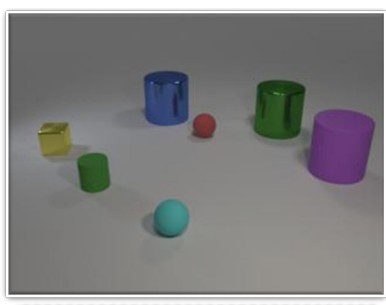 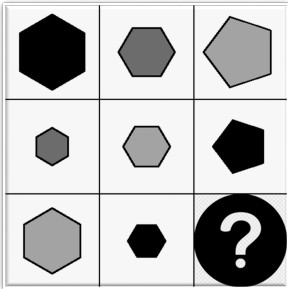 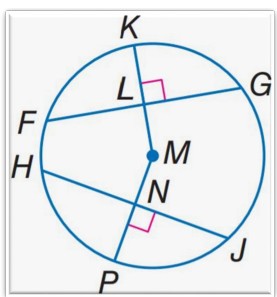

**Q:** Hint: Please answer the question requiring an integer answer and provide the final value, e.g., 1, 2, 3, at the end.\nQuestion: Subtract all red things. Subtract all tiny matte balls. How many objects are left?
**A:** 5

**Q:** Can it be affirmed that this image logically concludes the given sequence? Yes or no.

**A:** Yes.

**Q:** In $\odot M$, $FL=24, HJ=48$, and $m \widehat{HP}=65$. Find $m \widehat{HJ}$.\nChoices:\n(A) 65\n(B) 120\n(C) 130\n(D) 155"
**A:** A

Figure 28: Examples of math task in ability continual learning.

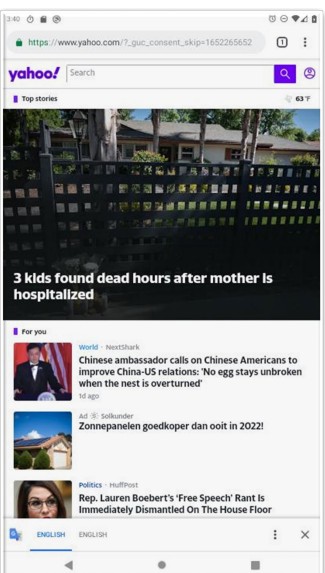

**Q:** You are an assistant in Android GUI navigation. You are given a screenshot image of an Android phone with the width and height of 412 and 732, respectively. The image size is scaled to [0,1] with the left upper point being [0,0]. My goal is \"toggle translation in the chrome app. Please select the most appropriate option to achieve my goal.
A. Click at position [0.18, 0.25]
B. Click at position [0.86, 0.92]
C. Click at position [0.91, 0.24]
D. Click at position [0.21, 0.83]
**A:** C

**Q:** You are an assistant in Android GUI navigation. You are given a screenshot image of an Android phone with the width and height of 412 and 732, respectively. The image size is scaled to [0,1] with the left upper point being [0,0].\nMy goal is \"turn off notifications settings in the gmail app. Please select the most appropriate option to achieve my goal.
A. Click at position [0.91, 0.24]
B. Click at position [0.44, 0.92]
C. Click at position [0.21, 0.83]
D. Click at position [0.28, 0.85]
**A:** D

**Q:** You are an assistant in Android GUI navigation. You are given a screenshot image of an Android phone with the width and height of 412 and 732, respectively. The image size is scaled to [0,1] with the left upper point being [0,0]. My goal is \"turn off picture-in-picture. Please select the most appropriate option to achieve my goal.
A. Click at position [0.18, 0.25]
B. Click at position [0.6, 0.34]
C. Click at position [0.91, 0.24]
D. Click at position [0.21, 0.83]
**A:** C

Figure 29: Examples of GUI agent task in ability continual learning.

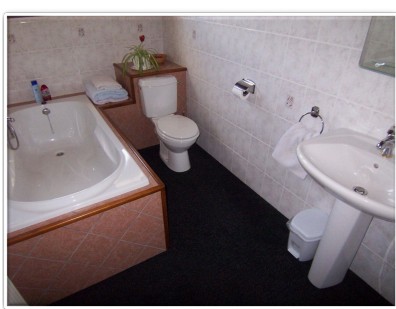 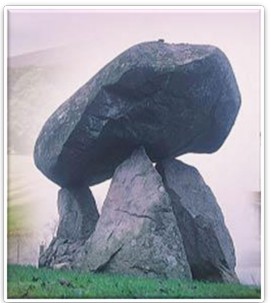 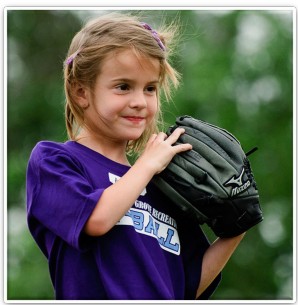

**Q:** Answer in natural language. How many mirrors are in the image? Choose between the following options: 2, 0, 3, or 1.
**A:** 1

**Q:** How many stones are in the image? Choose between the following options: 2, 4, 3, 5, 6, or 0.
**A:** 4

**Q:** How many baseball gloves are in the image? Choose between 3, 0, 2, or 1.
**A:** 1

Figure 30: Examples of visual perception task in ability continual learning.

## G.2 VISUALIZATION OF RESULTS

Fig. 31 provides examples during DCL and ACL, respectively. We can find that some baselines like LoRA (Hu et al., 2021), MoELoRA (Chen et al., 2024a), HiDE (Guo et al., 2025a) overfit to the last learned task and output options that do not exist in domain continual learning. In ACL, most baselines, including HiDE (Guo et al., 2025a), DISCO (Guo et al., 2025b), CL-MoE (Huai et al., 2025), etc., miss part of their OCR ability and do not answer the question correctly.

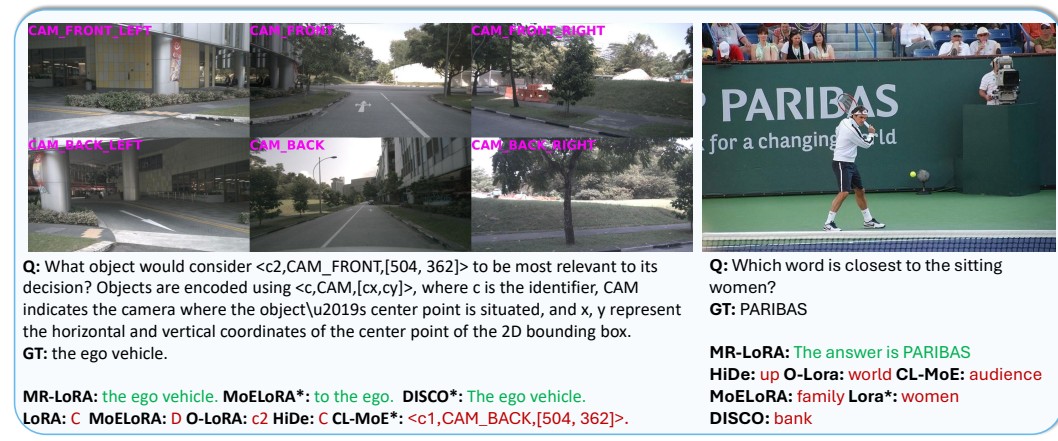

**Q:** What object would consider <c2,CAM_FRONT,[504, 362]> to be most relevant to its decision? Objects are encoded using <c,CAM,[cx,cy]>, where c is the identifier, CAM indicates the camera where the object's center point is situated, and x, y represent the horizontal and vertical coordinates of the center point of the 2D bounding box.
**GT:** the ego vehicle.

**MR-LoRA:** the ego vehicle. **MoELoRA\*:** to the ego. **DISCO\*:** The ego vehicle.
**LoRA:** C **MoELoRA:** D **O-LoRA:** c2 **HiDe:** C **CL-MoE\*:** <c1,CAM_BACK,[504, 362]>.

**Q:** Which word is closest to the sitting women?
**GT:** PARIBAS

**MR-LoRA:** The answer is PARIBAS
**HiDe:** up **O-Lora:** world **CL-MoE:** audience
**MoELoRA:** family **Lora\*:** women
**DISCO:** bank

Figure 31: Visualization of MR-LoRA and other baselines under domain continual learning and ability continual learning. The left part is testing the autonomous driving task after learning all domain tasks, while the right part is testing the OCR tasks after learning all ability tasks.

