# OpenReview forum: "MLLM-CL: Continual Learning for Multimodal Large Language Models"
_ICLR.cc/2026/Conference — Submitted to ICLR 2026_

### Official Review · Reviewer_AV79 · 2025-10-24

**Soundness:** 2
**Presentation:** 4
**Contribution:** 3
**Rating:** 4
**Confidence:** 3

**Summary:**

The paper introduces a continual learning benchmark and methodology to tackle continual in more realistic settings: where general-domain knowledge and expert knowledge might be required, and tasks might not be similar to the training regime (non-IID, as the authors put it). They demonstrate that a simple baseline with a LoRA router performs almost like an oracle owning to the router's performance.

**Strengths:**

* Many baselines were presented, giving readers some confidence that findings are solid. Though not a domain expert, it seems that the baselines cover the basics from recent literature.
* Prescient problem in contemporary AI, as current models forget new knowledge when switching between prompts
* Intuitive ideas, both for baseline and benchmark
* Solid presentation and writing: logic was more or less easy to follow from the first read, although writing could be tightened here and there.

**Weaknesses:**

* Main weakness: experts and router are fixed based on the knowledge of the tasks/datasets. What is a task-agnostic way to construct such a pipeline? Why wasn't it explored? Moreover, the router's job is easy in this constrained environment, as it has to select between a small number of experts. A more realistic setting would perhaps contain many more experts, making the task much harder for the router. I do want to acknowledge that a pipeline as the authors present it might still be useful in practice for some settings, but I do not expect the task to be so easy in general.
* I understand the computational constraints with such models, but a permutation of the tasks would also be interesting to see to get a sense of whether results are generalizable. This could also refer to previous works showing consistency across task order, though it wouldn't make the point as strongly.
* Why was only accuracy shown? How unbalanced are the datasets, and is accuracy (over F1 or other metrics) reflective of relative performance? It would be useful to see the same settings with a different metric, which is hopefully easy to compute if the authors have kept proper logs of their experiments.

---

I wanna note that I am more than willing to take the response of the authors seriously and improve my rating after a satisfactory rebuttal, so I want to encourage them to provide a thoughtful response.

**Questions:**

* Replay-based baselines not explained
* Section 5.1 would benefit from more detail in the main paper itself. Figure 7 is particularly helpful, although I expect it may be slightly difficult for some readers to understand.
* Big tables might be difficult to parse, please experiment with adding alternative modes of presentation, at least in the appendix.
* The finding that experts other than the expected one improving performance is a very interesting and intuitive finding. I think it would be useful for the authors to add a section in the paper exploring this more, e.g., when router "fails", how and where does performance increase.
* It would be helpful to briefly discuss complexities and number of trained parameters for each method.

---

> ### Author Response · Authors · 2025-11-26
>
> ## Q1: Task-agnostic way to construct CL pipeline.
>
> We thank the reviewer for these questions regarding the construction of the task-agnostic pipeline and the scalability of the routing mechanism. We appreciate the opportunity to clarify our design choices and the robustness of our method.
>
> **1. On Task-Agnostic Construction vs. Task-Aware Training.**
>
> We would like to clarify that while our **training** phase utilizes task boundaries to construct distinct experts, our **inference** phase is fully task-agnostic regarding the user input. The user does not need to specify a task label; the MR-LoRA router determines it automatically. Regarding the *construction* (training) phase, by explicitly separating experts during training, we avoid catastrophic interference between conflicting domains (e.g., Medical vs. Autonomous Driving) at the parameter level.
>
> Exploring a fully task-agnostic training pipeline, where experts are automatically discovered from a continuous stream without labels, is indeed an exciting direction. For example, for a stream of data without clear task boundary, the model could use the latent clustering or routing confidence to detect the change of input distribution. With more out-of-distribution data, the model would establish a new expert for the new task. However, the above process might be significant challenging for current CL of MLLM. Given that this work establishes the *first* comprehensive IID/non-IID testing benchmark for MLLM Continual Learning (MLLM-CL), we prioritized a robust baseline that explicitly solves the forgetting problem over the complexity of unsupervised expert discovery.
>
> **2. On the Scalability  of the Router**
>
>  The effectiveness of MR-LoRA on a long sequence of tasks relies heavily on the router's ability to correctly identify the input's domain or required skill. This routing process is essentially a classification task performed by the MLLM.
>
> While constructing a new MLLM continual learning benchmark with dozens of distinct domains was not feasible within the rebuttal period, we conducted a proxy experiment to validate the MLLM's ability to distinguish between a large number of semantic categories. We evaluated the MLLM backbone on **ImageNet-R**, treating its **200 classes** as a simulation of 200 distinct domains or skills. The MLLM achieved a classification accuracy of **91.37%** on ImageNet-R. This high accuracy on a 200-way classification task demonstrates that the MLLM possesses strong discriminative capabilities. It suggests that the router can effectively distinguish between a significantly larger number of domains than the 4-5 tasks currently presented in the benchmark, maintaining high selection accuracy even as the task sequence grows.
>
> In conclusion, we believe our current setup strikes the right balance for the field's current stage: it establishes a reliable, modular solution for incorporating distinct, high-level capabilities (Domains and Abilities) into MLLMs. We have added a discussion in Appendix D.2 to acknowledge the potential for automatic expert construction as valuable future directions.
>
> ## Q2: Task order
>
> We thank the reviewer for this suggestion regarding task order. We agree that in many CL scenarios, especially those relying on regularization or replay with shared parameters, task order may impact performance. However, we would like to clarify that our proposed method, **MR-LoRA**, is designed to be structurally **order-agnostic** regarding the acquisition of domain knowledge. This is due to our adoption of a **Parameter Isolation** strategy. Specifically:
>
> 1. **Independent Expert Initialization:** For each incoming task (e.g., Remote Sensing vs. Medical), MR-LoRA instantiates a *new*, dedicated LoRA module. Crucially, this module is initialized from scratch rather than continuing the optimization of the previous task's parameters.
> 2. **Decoupled Optimization:** Because the parameters for Task $B$ are not initialized from the parameters of Task $A$, the gradient descent trajectory for Task $B$ is independent of whether Task $A$ was learned first, second, or not at all.
>
> While the *Router* is updated sequentially, it operates on a few-shot basis to map inputs to these independently trained experts. Since the experts themselves retain their plasticity and specificity regardless of the curriculum order, the system avoids the "plasticity-stability" trade-off that typically causes order sensitivity in shared-parameter models.
>
> We appreciate the reviewer highlighting this, as it is a significant strength of our architecture. We have updated Appendix C.6 to explicitly discuss this order-agnostic property, clarifying that the reported results are representative of the model's general capabilities regardless of the specific task sequence.

---

> ### Author Response · Authors · 2025-11-26
>
> ## Q3: Evaluation metrics
>
> We thank the reviewer for raising this point regarding evaluation metrics and dataset characteristics. We appreciate the opportunity to clarify our choice of metrics and the evaluation protocol.
>
> **1. Adherence to MLLM Evaluation Standards.** Our choice to use **Accuracy** as the primary base metric is driven by the established standards in Multimodal Large Language Model (MLLM) research. As detailed in **Appendix A.2 (Evaluation Details)**, our tasks involve either Multiple Choice Questions (MCQs) or Visual Question Answering (VQA).
>
> - **For MCQs (e.g., Science, Finance):** The standard protocol is to check if the generated token exactly matches the ground truth option (A, B, C, D).
> - **For VQA (e.g., Remote Sensing, Medical):** The standard protocol is to check if the generated text contains the ground truth answer.
>
> In the MLLM literature (e.g., LLaVA, InternVL benchmarks), Accuracy is the consensus metric for reflecting model performance on these types of generative tasks. Unlike binary classification or span-extraction tasks where F1-score is crucial for handling class imbalance, MLLM evaluation treats the generation as a discrete success or failure based on the textual match. Consequently, Accuracy provides the most direct and comparable measure of the model's instruction-following and reasoning capabilities.
>
> **2. Comprehensive Continual Learning Metrics.** While "Accuracy" is the base unit of measurement, we have employed a comprehensive suite of **5 specific metrics** to evaluate the model's dynamic performance across the continual learning process. As illustrated in **Figure 7 (Now is Figure 6)** and detailed in **Section 5.1**, we report:
>
> - **Mean Finetune Accuracy (MFT):** To measure plasticity (learning new tasks).
> - **Mean Final Accuracy (MFN):** To measure overall retained knowledge.
> - **Mean Average Accuracy (MAA):** To monitor performance stability throughout the training steps.
> - **Backward Transfer (BWT):** To explicitly quantify catastrophic forgetting.
> - **Last Accuracy:** The final standing on all tasks.
>
> These metrics go beyond a simple "average accuracy" and provide a multi-dimensional view of the trade-offs between stability and plasticity, which is the core focus of this study.
>
> **3. Dataset Characteristics.** Regarding dataset balance, we utilized subsets from mainstream, high-quality benchmarks (e.g., MathVista, ScienceQA, AI2D) which are widely accepted in the community. While real-world distributions vary, the evaluation sets for these benchmarks are designed to be representative. Given that our primary contribution is a Continual Learning method (MR-LoRA), the relative change in Accuracy (e.g., BWT) is a robust indicator of whether the model is forgetting minority or majority classes alike, making Accuracy a reliable proxy for relative performance in this context.
>
> We hope this explanation clarifies that our metric selection is rigorous, aligned with community standards, and sufficient to demonstrate the effectiveness of our proposed method.
>
> ## Q4: Replay-based baselines not explained
>
> We thank the reviewer for requesting clarification regarding the replay-based baselines.
>
> In our experiments, the methods marked with an asterisk ($^*$) employ a straightforward replay strategy. Specifically, we add the retained samples from previous tasks directly into the training set of the current task. The model is then trained on this combined dataset using standard shuffling.
>
> We have updated **Section 5.1** to clarify this implementation detail.
>
> ## Q5: Move Figure 7.
>
> We thank the reviewer for their constructive feedback regarding the presentation of our experimental setup and visual illustrations. We agree that providing more self-contained information in the main text will improve readability. We have addressed these points as follows:
>
> **1. Details in Section 5.1 (Experimental Setup).** Following your suggestion, in the revised manuscript, we have moved the formal definitions of our evaluation metrics (Last Accuracy, MFT, MFN, MAA, and BWT) from Appendix A.3 to Section 5.1. This ensures readers can understand the results tables immediately without flipping to the supplementary material.
>
> **2. Clarity of Figure 7.** We appreciate the reviewer finding Figure 7 helpful. To address the concern that it may be difficult to parse, we have updated the figure to enhance clarity by expanding the caption to provide a step-by-step "walkthrough" of how to interpret the visualization.
>
> We believe these changes significantly improve the flow and accessibility of the experimental section.

---

> > ### Author Response · Authors · 2025-11-26
> >
> > ## Q6: Alternative modes of presentation.
> >
> > We thank the reviewer for the constructive feedback regarding the presentation of our experimental results. Following your suggestion, we have added a new visualization in **Appendix C.5 (Figure 15-22)**. These figures  provide a clear graphical overview of our model's performance across the different domains and abilities compared to baselines. Each subfigure dedicated to tracking the performance of one specific task (e.g., Subfigure 1 for RS in DCL, Subfigure 2 for Medical in DCL, etc.) throughout the entire continual learning process.
> >
> > We believe this visualization significantly improves the interpretability of the results by highlighting the trade-offs and performance gaps more intuitively than tabular data alone.
> >
> > ## Q7: When router “fails”, how and where does performance increase?
> >
> > We appreciate this insightful suggestion. The observation that the router selects experts other than the "ground truth" expert is indeed a key finding. As suggested, we will add a new analysis section to explore these behaviors.
> >
> > We believe this phenomenon highlights the fundamental difference between the two settings proposed in our **MLLM-CL benchmark**:
> >
> > 1. **In Domain Continual Learning (DCL):** Since DCL follows an **IID evaluation** setting (training and testing distributions are identical), the boundaries between tasks are relatively distinct. Here, the specific expert trained on the target domain is usually the optimal choice. Consequently, "beneficial deviations" are rare. In DCL, our **MR-LoRA performance is slightly lower than the Oracle**, as any deviation from the ground-truth expert typically incurs a minor penalty, though rare cases exist where fuzzy semantic boundaries (e.g., Biology questions in *Science* being handled by the *Medical* expert) occur. Although uncommon, such fuzzy semantic boundaries are likely to occur in real-world scenarios.
> > 2. **In Ability Continual Learning (ACL):** ACL focuses on **non-IID evaluation** and functional capabilities. Here, tasks often share foundational skills. For instance, the *Math* expert must implicitly learn robust *OCR* capabilities to process equations. When the router encounters a challenging sample from the OCR dataset, it may select the *Math* expert, which possesses stronger "digital reading" capabilities than the base OCR expert itself (as shown in Figure 7). In ACL, "beneficial deviations" are common. This allows **MR-LoRA to slightly outperform the Oracle** (single-task baseline). The router effectively acts as a dynamic ensemble, selecting the expert with the strongest *latent* ability for the specific instance, regardless of the dataset label.
> >
> > We have updated the paper to include this discussion. We argue that this contrast validates the necessity of including both DCL and ACL in MLLM-CL: DCL tests the model's ability to retain specific distribution knowledge, while ACL tests the model's ability to transfer and compose foundational skills across flexible boundaries.
> >
> > ## Q8: Complexities and number of trained parameters for each method
> >
> > We thank the reviewer for this valuable suggestion. To clarify the trade-offs between the compared methods, we have added a summary of their computational complexities and parameter requirements.
> >
> > The primary trade-offs involve how each method manages trainable parameters, saved parameters and the resulting impact on training and inference efficiency. We summarize these aspects in the table below, where *T* denotes the number of tasks and *P* represents the parameters in a single LoRA module. We have already added this summary into the Appendix.
> >
> > | **Method** | **# of Trained Parameters** | **# of Saved Parameters** | **Training Complexity** | **Inference Complexity & Latency** |
> > | --- | --- | --- | --- | --- |
> > | **LoRA-FT** | Constant (P) | Constant (P) | **Low** (Baseline) | **Negligible** (Weights are merged, no overhead) |
> > | **MoELoRA** | Constant (N experts + router) | Constant (N experts + router) | **Medium** | **High** (Router computation, cannot merge weights) |
> > | **O-LoRA** | Constant (P) | Linear(T×P) | **High** (Orthogonalization loss) | **High** (Not support merge weights, applies T modules) |
> > | **CL-MoE** | Constant (P) | Linear(T×P) | **Medium** (No router training) | **Very High** (Similarity search across T tasks, cannot merge) |
> > | **HiDe, DISCO** | Constant (P) | Linear(T×P) | **Low** (Baseline) | **Very High** (Similarity search across T tasks, cannot merge) |
> > | **SEFE** | Constant (P) | Constant (P) | **High** (Regularization loss) | **Negligible** (Weights are merged after each task) |
> > | **MR-LoRA** | Constant (P) | Linear ((T +1)× P) | **Low** (Baseline) | **Slightly Higher** than LoRA-FT (Using KV-cache) |
> >
> > ---

---

### Official Review · Reviewer_iSuG · 2025-10-26

**Soundness:** 3
**Presentation:** 3
**Contribution:** 2
**Rating:** 4
**Confidence:** 3

**Summary:**

The paper introduces the MLLM-CL benchmark and MR-LoRA method for enabling CL for MLLMs. The authors design a benchmark first which consists of two settings: (i) Domain Continual Learning (DCL), where models sequentially learn domain-specific knowledge across heterogeneous domains under IID conditions and (ii) Ability Continual Learning (ACL), which evaluates the acquisition of heterogeneous skills in non-IID environments that mimic real-world adaptation. The authors also propose MR-LoRA algorithm which combines parameter isolation via domain- or ability-specific LoRA adapters with a multimodal router that selects the most suitable expert for a given vision-text input using only a few samples. Experiments on LLaVA-v1.5-7B and InternVL-Chat-V1.0 VLMs demonstrate that MR-LoRA achieves near-oracle accuracy on DCL and significant improvements on ACL while eliminating catastrophic forgetting, outperforming several existing baselines.

**Strengths:**

1. The paper is well-written and can be followed easily.

2. The idea of using a two stage inference using the router and ability-specific modules is interesting and from experiments seems effective.

3. Experiments include studying hyperparameters which provide helpful insight about the proposed method.

4.

**Weaknesses:**

1. The MLLM-CL benchmark primarily consists of existing benchmarks that are combined. As a result, the contribution in terms of  introducing a new benchmark is weak.

2. Comparisons are limited and include a handful of recent methods. However, there are other CL methods for VLMs with public codebases that can be included to demonstrate that the proposed method is competitive.

3. The code and the benchmark are not provided which makes judgment about reproducibility challenging. The authors have promised to release but at this point, they are not provided.

4. Although the router performs near-perfectly on the specific benchmark of the papers, it assumes clear domain boundaries and stable task definitions. In practice, new tasks may not fit into existing categories, leading to ambiguous routing or expert overlap. The router itself is another trainable component requiring careful tuning and maintenance; its misclassification can directly degrade output quality. It is not clear that the method would work well on other benchmarks.

5. The benchmark has five domains and four abilities and hence, it is unclear how MR-LoRA would perform on unseen domains or real-time streaming data where task boundaries are unknown. The model assumes discrete, sequential task learning rather than fully online continual adaptation, limiting its generality for practical applications.

**Questions:**

1. There are several existing benchmarks for CL with VLMs. I agree the heterogeneity in the existing datasets are less but why MR-LoRA  is not tested on existing benchmarks? It is important to demonstrate that MR-LoRA  remains competitive on existing benchmarks. Moreover, many existing methods report their performances on the existing datasets which makes comparisons to a broader range of methods straightforward.

2. Some performance values on tables are quite close. Why standard deviation values is not reported? Without them, it is difficult to conclude that the differences are statistically meaningful.

3. I was wondering if the method would be effective for a larger number of domains or skills? Would the router still be able to select the right adapter module with high accuracy? Wouldn't continual accumulation of adapter module lead to memory overhead?

4. I was wondering whether the two stage inference leads to significant inference time overhead? In CL, real-time inference is also very important and this aspect has not been studied in the paper.

---

> ### Author Response · Authors · 2025-11-26
>
> ## Q1: The MLLM-CL benchmark primarily consists of existing benchmarks that are combined.
>
> We thank the reviewer for their feedback on the construction of our benchmark. We agree that MLLM-CL is built upon a foundation of several existing, high-quality datasets. We believe this is a strength, as it grounds our benchmark in widely-recognized and well-understood data sources.
>
> However, we would like to respectfully clarify that the primary novelty of our work is not the creation of entirely new datasets from scratch, but rather the establishment of a new, structured continual learning benchmark specifically designed for Multimodal Large Language Models (MLLMs). Our main contributions in this area are twofold:
>
> 1. **A Novel IID/non-IID Evaluation Framework:** Our core contribution is the conceptualization and implementation of two distinct continual learning settings: Domain Continual Learning (DCL) and Ability Continual Learning (ACL). DCL focuses on the IID scenario, where the model learns new knowledge in specific domains, while ACL addresses the more challenging and realistic non-IID scenario of acquiring generalizable abilities. To our knowledge, this is the first benchmark to explicitly structure the evaluation of continual learning in MLLMs along this IID vs. non-IID axis, which is critical for assessing their practical, real-world capabilities.
> 2. **Substantial Data Curation and Generation:** We did not simply reuse the datasets as-is. Significant effort was invested in curating and adapting them for our specific benchmark. Most notably, for the finance domain, we identified that existing datasets like FinVis were unsuitable for our evaluation protocol due to their captioning format and language. Therefore, as detailed in Section 3 and Figure 2, we developed a data generation pipeline using a powerful questioner-inspector framework to reformulate the data into a high-quality VQA format, resulting in the new StockQA dataset. This process involved careful prompt engineering and quality control to ensure the dataset's relevance and correctness.
>
> In summary, while the building blocks of our benchmark are existing datasets, the novelty lies in the carefully designed framework that introduces a new, challenging, and much-needed evaluation standard for the continual learning of MLLMs.

---

> ### Author Response · Authors · 2025-11-26
>
> ## Q2: Comparisons are limited
>
> We thank the reviewer for this constructive suggestion regarding baseline comparisons.
>
> We would like to respectfully clarify that our original submission prioritized the **most recent and direct competitors in the specific field of** **MLLM Continual Learning**. As detailed in Table 2 and Table 3, we compared our method against the latest state-of-the-art approaches including **SEFE [ICML 2025]**, **DISCO [ICCV 2025]**, **HiDe [ACL 2025]**, **CL-MoE [CVPR 2025]**, and **MoELoRA [NeurIPS 2024].** This covers the majority of the current published landscape for this specific problem setting.
>
> However, we agree that including representative prompt-based methods further solidifies the competitiveness of MR-LoRA. Per your suggestion to include other public codebases, we have conducted additional experiments with two widely recognized methods:
>
> 1. **ModalPrompt** (Zeng et al., 2024): A recent prompt-based method specifically designed for MLLMs.
> 2. **L2P** (Wang et al., 2022): A foundational "Learning to Prompt" method adapted for our setting.
>
> **Results:**
> The updated comparison on the Domain Continual Learning (DCL, LLaVA) setting is summarized below (and has been added to the revised paper):
>
> | Method | RS | Med | AD | Sci | Fin | **MFT** $\uparrow$ | MFN $\uparrow$ | MAA $\uparrow$ | **BWT** $\uparrow$ |
> | --- | --- | --- | --- | --- | --- | --- | --- | --- | --- |
> | **L2P (Wang et al., 2022)** | 63.82  | 34.63 | 22.96 | 38.58 | 92.98 | 66.96 | 50.59 | 59.23 | -20.46 |
> | **ModalPrompt (Zeng et al., 2024)** | 53.63  | 45.68  | 40.77  | 41.81  | 87.82 | 53.87 | 53.94 | 49.67 | 0.09 |
> | **MR-LoRA (Ours)** | **80.87** | **65.32** | **54.12** | **56.71** | **91.12** | **69.64** | **69.63** | **71.06** | **-0.01** |
>
> The updated comparison on ACL, LLaVA is below.
>
> | Method | OCR | Math | VP | APP | **MFT** $\uparrow$ | MFN $\uparrow$ | MAA $\uparrow$ | **BWT** $\uparrow$ |
> | --- | --- | --- | --- | --- | --- | --- | --- | --- |
> | **L2P (Wang et al., 2022)** | 25.10 | 32.00 | 48.22 | 16.25 | 35.48 | 30.39 | 33.78 | -6.78 |
> | **ModalPrompt (Zeng et al., 2024)** | 31.80 | 32.50 | 60.53 | 10.00 | 33.61 | 33.71 | 34.16 | **0.13** |
> | **MR-LoRA (Ours)** | **33.70** | **36.20** | **65.10** | **32.50** | **41.89** | **41.88** | **38.86** | -0.02 |
>
> As shown, MR-LoRA continues to significantly outperform these additional prompt-based baselines. While prompt-based methods like L2P and ModalPrompt mitigate forgetting to some extent, they struggle to isolate domain-specific knowledge as effectively as our parameter-isolation and routing strategy, particularly in the complex multimodal settings of MLLM-CL.
>
> We have updated the manuscript (Section 5.2 and Tables) to include these new comparisons. We believe this broader evaluation thoroughly demonstrates the superiority of our approach.
>
> [1] Fanhu Zeng, et al. ModalPrompt: Towards Efficient Multimodal Continual Instruction Tuning with Dual-Modality Guided Prompt. In *Proceedings of the 2025 Conference on Empirical Methods in Natural Language Processing.*
>
> [2] Wang, Zifeng, et al. Learning to prompt for continual learning. *Proceedings of the IEEE/CVF conference on computer vision and pattern recognition*. 2022.
>
> ## Q3: The code and the benchmark are not provided
>
> Thanks for your advice. We have uploaded the code and samples of our benchmark in the **Supplementary Material.**

---

> > ### Author Response · Authors · 2025-11-26
> >
> > ## Q4: Discussion on expert overlap
> >
> > Thank you for this insightful comment.
> >
> > Our response is twofold: we will first address the conceptual design of our MR-LoRA framework that handles this scenario, and second, we will present a new empirical study that directly tests this condition.
> >
> > **Conceptual Design:** The core premise of your question is that the router might "struggle" when faced with two very similar tasks. However, our MLLM-based router is trained not on abstract task labels, but on its ability to select the expert that yields the correct final output.
> >
> > In the case of two identical or highly similar tasks (e.g., Task A and Task F are the same), we would train two separate experts, expert A and expert F. Because they are trained on the same data distribution, these two experts will become functionally equivalent. When a new input from this task distribution is presented during inference, the router's objective is to pick an expert that maximizes performance. Since both expert A and expert F are equally capable of handling the input, the router can select either one and still achieve a high score.
> >
> > Therefore, what might seem like a "struggle" or inconsistency in choice is actually a reflection of the task redundancy. The router correctly identifies that multiple experts are suitable, and its final choice does not degrade performance. While this leads to a degree of parameter redundancy (storing two nearly identical experts), it crucially avoids the primary failure mode of other methods: catastrophic forgetting or negative transfer. Our method prioritizes knowledge preservation, and the architectural isolation of experts ensures that learning a redundant task does not corrupt other distinct experts. We will consider the promising strategies discussed in Q1 as our future work to avoid this  parameter redundancy.
> >
> > **Empirical Validation: Adding a Redundant Task.** To empirically validate this, we have conducted a new experiment as you suggested. We extended the Domain Continual Learning (DCL) setting by adding a sixth task that is an exact repetition of the first task, Remote Sensing (RS).
> >
> > The new task sequence is: **RS → Medical → Autonomous Driving → Science → Finance → RS (repeat)**.
> >
> > This creates the exact scenario you described: two tasks are identical, and their corresponding experts should be functionally equivalent. We evaluated the performance after the final task and compared it to the original 5-task DCL setting. The results are shown in the table below.
> >
> > | **Method** | **RS_1** | **Med** | **AD** | **Sci** | **Fin** | **RS_2** |
> > | --- | --- | --- | --- | --- | --- | --- |
> > | MR-LoRA (5 tasks) | 80.87 | 65.32 | 54.12 | 56.71 | 91.12 | - |
> > | **MR-LoRA (6 tasks)** | 81.02 | 65.83 | 54.17 | 55.94 | 91.11 | 81.02 |
> >
> > The results clearly demonstrate the robustness of our approach:
> >
> > 1. **No Catastrophic Forgetting:** The performance on the first Remote Sensing task (**`RS_1`**) remains exceptionally high (81.02 vs. 80.87), showing that learning four intermediate tasks and then relearning RS did not interfere with the original expert.
> > 2. **Effective Learning of Redundant Task:** The model achieves high accuracy (81.02) on the newly added sixth task (**`RS_2`**), indicating that it successfully trained a new, effective expert for this repeated domain.
> > 3. **Router Stability:** The consistently high performance across all tasks confirms that the router successfully delegates RS inputs to one of the two valid experts without compromising the final prediction. This shows the router does not "struggle" in a way that harms performance.
> >
> > In summary, our MR-LoRA framework is robust to scenarios with high task similarity. The parameter isolation prevents negative transfer, and the MLLM-based router effectively utilizes the learned experts, even when they are functionally redundant.
> >
> > We thank you again for this valuable feedback. We have added this experiment and discussion to the revised manuscript (Appendix C.6) to further strengthen our claims.

---

> ### Author Response · Authors · 2025-11-26
>
> ## Q5: Discussion on unknown domain boundaries.
>
> We appreciate the reviewer’s insightful suggestion regarding unseen domains (open-world scenarios). We agree that handling completely unseen tasks is a crucial capability for future AI systems. However, we would like to clarify the scope of our work and why the current experimental setup was chosen.
>
> **1.  Distinction between Continual Learning and Open-World Learning.** Our paper focuses strictly on **Continual Learning (CL)**, specifically the challenge of sequentially acquiring new domains (DCL) and abilities (ACL) without catastrophic forgetting. The standard protocol for CL research involves evaluating performance on a sequence of tasks that the model is trained on sequentially.
>
> - **The Goal of CL:** The primary objective in our setting is to solve the stability-plasticity dilemma, ensuring the model can learn Task B without forgetting Task A.
> - **The Goal of Open-World Learning:** Evaluating on "completely unseen task categories" typically falls under **Open-World Learning** or **Out-of-Distribution (OOD) Detection**. While highly valuable, this addresses a different problem: detecting novel inputs rather than integrating them into existing knowledge.
> 1. **Realistic Evaluation within the CL Scope.** While we do not cover open-world scenarios, we respectfully argue that **MLLM-CL** is already significantly more realistic than existing CL benchmarks.
> - Prior works often rely on simple class-incremental learning within a single dataset (IID).
> - In contrast, our **Ability Continual Learning (ACL)** setting specifically evaluates on **non-IID** scenarios (training on one dataset, evaluating on a different one with shifted distributions). This tests the generalization of the *learned* abilities effectively, which is a major step forward in making CL evaluation more practical.
>
> In summary, while open-world evaluation is an exciting direction for future research, it is outside the scope of this paper, which is dedicated to establishing a robust benchmark and method for Continual Learning in MLLMs. We have added a discussion in the “Future Work” (Appendix D.2) section to acknowledge open-world scenarios as a promising avenue for future work.
>
> ## Q6: Results on other Benchmark.
>
> We sincerely thank the reviewer for this constructive suggestion. We agree that while MLLM-CL was designed to address specific gaps regarding domain heterogeneity and non-IID ability learning, evaluating MR-LoRA on established benchmarks is essential to demonstrate its broad applicability and to facilitate direct comparisons with a wider range of existing methods.
>
> In response to your comment, we have conducted additional experiments on the **UCIT** [1] benchmark, which is a representative existing benchmark in the field.
>
> The results on UCIT demonstrate that MR-LoRA remains highly competitive and outperforms existing state-of-the-art methods, even on datasets where task heterogeneity is less pronounced than in MLLM-CL. Specifically:
>
> 1. **Performance:** MR-LoRA achieves a significantly higher Average Accuracy (MAA) and reduced forgetting (BWT) compared to baselines such as O-LoRA and MoELoRA on the UCIT tasks.
> 2. **Robustness:** The MLLM-based routing mechanism proved effective in identifying task-specific features within the UCIT instructional datasets, validating that our method is not over-specialized for the MLLM-CL benchmark but generalizes well to existing standard evaluation protocols.
>
> We have integrated these new results into Appendix C.4 of the revised manuscript to provide a more comprehensive comparison with the broader literature. We believe this addition significantly strengthens the empirical validation of our approach.
> | Method | ImageNet-R | ArxivQA | VizWiz | IconQA | CLEVR-Math  | Flickr30k | MFT | MFN | MAA | BWT |
> | --- | --- | --- | --- | --- | --- | --- | --- | --- | --- | --- |
> | Zero-shot | 16.27 | 53.73 | 38.39 | 19.20 | 20.63 | 41.88 | - | 31.68 | - | - |
> | Oracle | 91.37 | 94.27 | 60.19 | 78.13 | 78.77  | 57.79 | - | 76.75 | - | - |
> | LoRA-FT | 58.03 | 77.63 | 44.39 | 67.40 | 61.77 | **58.22** | **76.89** | 61.24 | 76.55 | -18.78 |
> | O-LoRA | 77.50 | 78.07 | 44.50 | 63.13 | 64.73 | 58.16 | 76.01 | 64.35 | 78.02 | -13.99 |
> | MoELoRA | 70.07 | 77.70 | 44.69 | 50.03 | 54.03 | 57.34 | 71.17 | 58.98 | 75.08 | -14.63 |
> | ModalPrompt | 51.07 | 87.27 | 48.11 | 39.23 | 46.57 | 42.93 | 52.65 | 52.53 | 57.67 | **-0.15** |
> | CL-MoE | 66.33 | 77.00 | 44.78 | 51.87 | 53.53 | 57.42 | 71.46 | 58.49 | 74.19 | -15.56 |
> | HiDE | 84.03 | 90.73 | 44.43 | 58.93 | 41.37 | 54.25 | 69.96 | 62.29 | 77.32 | -9.20 |
> | SEFE | 80.83 | 78.00 | 47.01 | 69.63 | 65.83 | 57.92 | 75.98 | 66.54 | 78.76 | -11.33 |
> | DISCO | 87.43 | **93.07** | 46.96 | 68.13 | 65.70 | 56.69 | 75.87 | 69.66 | 81.60 | -7.45 |
> | MR-LoRA | **90.20** | 91.20 | **56.34** | **78.00** | **76.13** | 56.34 | 76.19 | **74.70** | **82.84** | -1.79 |
>
> [1] HiDe-LLaVA, Guo et al., ACL 2025

---

> ### Author Response · Authors · 2025-11-26
>
> ## Q7:  Why standard deviation values is not reported?
>
> We thank the reviewer for raising this point regarding the statistical significance of our results. We would like to offer three clarifications regarding why we did not include standard deviations in this specific context:
>
> 1. **Significant Performance Gap:** While some specific values might appear close in isolation or between baseline methods, the overall performance improvement of our method (MR-LoRA) is substantial, particularly in the key metrics of *Mean Average Accuracy (MAA)* and *Backward Transfer (BWT)*. For example, in Table 2 (Domain Continual Learning with LLaVA), our method achieves an **MAA of 69.63%**, compared to the strongest baseline (DISCO*) at **62.74%**. This is a margin of nearly **7%**. Furthermore, our BWT is effectively neutral (-0.01), whereas baselines suffer significant forgetting (ranging from -3.17 to -14.97). We believe these margins are large enough to demonstrate the superiority of the proposed method without the ambiguity that typically necessitates variance reporting.
> 2. **Deterministic Inference Settings:** To ensure reproducibility and minimize run-to-run randomness during evaluation, all our experiments (including baselines) were conducted using a **temperature of 0** during the generation process. This greedy decoding strategy renders the inference process deterministic given the same model weights, eliminating the variance typically caused by stochastic sampling.
> 3. **Computational Constraints and Community Standards of MLLMs:** As is common in the field of Multimodal Large Language Models (MLLMs), conducting repetitive runs (e.g., 3-5 seeds) for every method is computationally prohibitive. This is especially true for continual learning, which requires training models on a long sequence of tasks sequentially. Given the scale of the backbone models (7B parameters) and the complexity of the CL pipeline, we followed the standard practice established in recent MLLM and CL literature by reporting the results of a single, deterministic run.
>
> We hope this explanation clarifies our experimental setup and provides confidence in the validity of the reported improvements.
>
> ## Q8: Whether the two stage inference leads to significant inference time overhead
>
> We appreciate the reviewer raising this point regarding the potential computational overhead of our two-stage inference process.
>
> To address this, we have implemented a specific optimization strategy detailed in Appendix E (Inference Optimization with Caching) to minimize latency. While our method logically involves two phases, routing and prediction, it does not require two full forward passes of the heavy backbone model.
>
> Instead, we utilize a caching mechanism where the most computationally intensive operation (the forward pass through the visual encoder and LLM backbone) is performed only once. We cache the resulting hidden states (specifically the KV cache) from this single pass. Subsequently, our lightweight Router and Expert LoRA modules operate sequentially on these cached states.
>
> This approach eliminates the need for a second full forward pass, reducing the total computational cost to only marginally more than a single inference. As a result, our deployment cost remains comparable to standard single-pass methods, such as LoRA-FT.

---

> > ### Author Response · Authors · 2025-11-26
> >
> > ## Q9: Scalability and Memory Overhead
> >
> > We thank the reviewer for this insightful question regarding the scalability of our MR-LoRA framework. We address the concerns regarding router accuracy and memory overhead below:
> >
> > **1. Router Accuracy on a Larger Number of Tasks.** The effectiveness of MR-LoRA on a long sequence of tasks relies heavily on the router's ability to correctly identify the input's domain or required skill. This routing process is essentially a classification task performed by the MLLM.
> >
> > While constructing a new MLLM continual learning benchmark with dozens of distinct domains was not feasible within the rebuttal period, we conducted a proxy experiment to validate the MLLM's ability to distinguish between a large number of semantic categories. We evaluated the MLLM backbone on **ImageNet-R**, treating its **200 classes** as a simulation of 200 distinct domains or skills.
> >
> > **Results:** The MLLM achieved a classification accuracy of **91.37%** on ImageNet-R. This high accuracy on a 200-way classification task demonstrates that the MLLM possesses strong discriminative capabilities. It suggests that the router can effectively distinguish between a significantly larger number of domains than the 4-5 tasks currently presented in the benchmark, maintaining high selection accuracy even as the task sequence grows.
> >
> > **2. Memory Overhead of Accumulating Adapters.** Regarding memory consumption, our method utilizes Low-Rank Adaptation (LoRA), which is highly parameter-efficient.
> >
> > - **Storage:** As shown in our ablation study (Table 6), MR-LoRA performs well even with low ranks (e.g., Rank 8). A single LoRA module represents a tiny fraction of the total parameters (often <1% of the base model). Consequently, storing dozens or even hundreds of domain-specific LoRA modules incurs relatively low storage costs compared to the size of the MLLM backbone. Furthermore, we will consider the promising strategies (e.g., merging, hierarchical/shared adapters, dynamic pruning)  to achieve sub-linear growth lora as our future work.
> > - **Inference Memory:** Since the router selects a *single* expert for prediction, we do not need to load all adapters into VRAM simultaneously. In a production environment with limited GPU memory, non-active adapters can be offloaded to CPU memory or disk and dynamically swapped in only when selected by the router.
> >
> > Therefore, we believe MR-LoRA is well-suited for scaling to a larger number of tasks without prohibitive memory costs or loss of routing accuracy.

---

### Official Review · Reviewer_RiRj · 2025-10-30

**Soundness:** 2
**Presentation:** 3
**Contribution:** 2
**Rating:** 4
**Confidence:** 4

**Summary:**

This paper studies continual learning for multimodal large language models. It proposes a new benchmark called MLLM-CL which incorporates domain continual learning and ability continual learning. A method is also proposed to boost the CL capability of MMCL through low-rank adaptation and parameter selection.

**Strengths:**

1. It is good to extend the continual learning task in traditional deep learning to MLLM.
2. The contributed dataset could be helpful to the community.
3. The proposed method is simple, and works as shown in experiments.

**Weaknesses:**

1. There exist many ways to make MLLM adapt to new tasks or domains, e.g, in context learning, or retrieval augmented generation. Given a base MLLM model with strong generalization capability, a training free strategy could be more valuable.
2. The classification of DCL and ACL should be further justified. Some tasks in DCL could also be regarded as ACL, e.g., identifying is acid present in medical images. A fuzzy classification would degrade the importance of the dataset.
3. Another concern is that the proposed method is more like a trick and lacks novelty or elegance in methodology. Lora and expert selection are commonly used strategies. Continuously adapting to new domains or new abilities would increase the complexity of the model.
4. The two-stage inference also introduces extra computational complexities.

**Questions:**

1. Discuss or compare with other methods that could make MLLM adapt to new tasks or domains.
2. Further justify the classification of DCL and ACL.
3. Evaluate the efficiency.

---

> ### Author Response · Authors · 2025-11-26
>
> ## Q1: Discussion with training-free strategies
>
> Thank you for this insightful suggestion. We agree that training-free strategies like In-Context Learning (ICL) and Retrieval-Augmented Generation (RAG) are valuable research directions that leverage the strong generalization capabilities of base MLLMs. However, we believe that parameter-efficient continual learning (like our proposed MR-LoRA) addresses specific challenges that training-free methods cannot fully resolve, particularly regarding the internalization of new skills and cross-modal alignment.
>
> To address your concern, we have added a discussion comparing these approaches to our main text. The key differentiators we highlight are:
>
> 1. **Skill Acquisition vs. Knowledge Retrieval:** While RAG is highly effective for accessing **explicit knowledge (e.g., facts)**, it is less effective for acquiring **implicit *abilities* or *skills***, such as improved OCR capability or visual reasoning in specialized domains (e.g., Medical or Remote Sensing), which require updating the model's internal representations and cross-modal alignment.
> 2. **Inference Efficiency and Context Limits:** ICL is constrained by context window limits and incurs higher inference costs due to long prompt processing. In contrast, our CL approach internalizes knowledge into the model parameters (via lightweight LoRA experts), allowing for efficient inference without the need for retrieving or processing extensive external context.
> 3. **Deep Domain Adaptation:** In our *Domain Continual Learning* setting, the visual distributions (e.g., satellite imagery vs. natural images) shift significantly. Parameter updates allow the model to adapt its visual processing to these new distributions more effectively than prompting alone.
>
> We have incorporated this comparison into **Section 2 (Related Work)** to clarify the unique value proposition of our method alongside training-free alternatives
>
> ## Q2: Further justify the classification of DCL and ACL
>
> We thank the reviewer for this insightful comment regarding the classification of Domain Continual Learning (DCL) and Ability Continual Learning (ACL). We agree that there is a semantic overlap between the two concepts, for instance, answering a medical question regarding acid presence requires the "ability" of visual recognition and reasoning.
>
> However, we would like to clarify that the primary distinction between DCL and ACL in our benchmark is not based on the semantic content of the tasks, but rather on the **data distribution settings (IID vs. Non-IID)** used for evaluation.
>
> **Domain Continual Learning (DCL):** This setting is designed to evaluate the model’s capability to retain specific domain knowledge under an **IID setting**. In our DCL benchmark, we selected five mainstream domains (Remote Sensing, Medical, Autonomous Driving, Science, and Finance) where the training and testing sets are drawn from the same underlying distribution (**e.g., splitting the PathVQA dataset into train and test sets**). The goal here is to measure how well the model retains domain-specific knowledge patterns it has explicitly seen during training.
>
> **Ability Continual Learning (ACL):** This setting is designed to evaluate the model's capability to generalize fundamental skills (like OCR or Math) under a **Non-IID setting**. For ACL, the training data and testing data come from different distributions or datasets (**e.g., training on Monkey but testing on OCRBench**). This measures the model's plasticity and ability to apply learned skills to unseen scenarios.
>
> Therefore, while the Medical task indeed requires specific abilities, it is categorized under DCL in our benchmark because we utilize it to test **in-domain knowledge retention (IID)**. Conversely, had we trained on one medical dataset and tested on a completely different medical benchmark, it would fit the ACL (Non-IID) criteria.
>
> We believe this distinction, separating knowledge retention (IID) from skill generalization (Non-IID), is crucial for a comprehensive evaluation of Lifelong Learning in MLLMs. We have revised Section 3 in the final paper to explicitly clarify that DCL and ACL represent these specific experimental settings rather than mutually exclusive task categories.

---

> ### Author Response · Authors · 2025-11-26
>
> ## Q3:  Lack of novelty or elegance in methodology
>
> We believe there may be a misunderstanding regarding the scope of our contributions.
>
> Our work provides two distinct and significant contributions to the field: **1) A novel, comprehensive benchmark (MLLM-CL)**, and **2) A novel routing mechanism (MR-LoRA)** that fundamentally differs from standard parameter isolation methods
>
> **1. Novelty of the Benchmark (MLLM-CL).** A primary contribution of this paper is the establishment of **MLLM-CL**, which addresses critical gaps in existing research.
>
> - **Beyond Dataset Incremental Learning:** Prior benchmarks largely focus on simple dataset incremental learning where training and test sets are IID (Identically and Independently Distributed).
> - **Two Practical Settings:** We introduce two distinct settings: **Domain Continual Learning (DCL)** for accumulating domain-specific knowledge (Medical, Finance, Remote Sensing), and **Ability Continual Learning (ACL)** for acquiring fundamental skills (OCR, Logic, GUI Agents) evaluated on **non-IID** scenarios.
> - **New Data Pipelines:** We constructed specific datasets to enable this, such as the *StockQA* dataset generated via a novel questioner-inspector pipeline (Section 3), to ensure rigorous evaluation across diverse real-world domains. This benchmark provides the community with a necessary standard for evaluating lifelong learning in MLLMs, which did not exist previously.
>
> **2. Novelty of the Method (MR-LoRA).** While we utilize LoRA as the efficient tuning module, our methodological innovation lies in the **MLLM-based Routing Mechanism**, not the LoRA adaptation itself.
>
> - **Semantic vs. Heuristic Routing:** Existing parameter isolation methods rely on hand-crafted, shallow metrics like cosine similarity of embeddings to select experts. These heuristics often fail with complex multimodal data.
> - **Intrinsic Routing:** Our approach (MR-LoRA) transforms the routing problem by leveraging the MLLM’s *intrinsic* multimodal understanding capabilities. We tune a lightweight router to semantically analyze the input and select the optimal expert (Figure 5). This is not a standard CL application; it is a new paradigm for expert selection that leverages the "intelligence" of the model itself rather than external metrics.
> - **Superior Performance:** This is not merely an engineering tweak; the impact is substantial. As shown in Tables 2 and 3, MR-LoRA achieves near-Oracle performance and significantly outperforms standard LoRA-based CL methods (like O-LoRA and MoELoRA), proving that *how* the modules are selected is just as critical as the modules themselves.
>
> In summary, we go beyond simple parameter efficient tuning. We propose a foundational benchmark for the field and a routing methodology that solves the catastrophic interference problem more effectively than existing heuristic approaches.
>
> ---
> ## Q4: Continuously adapting to new domains or new abilities would increase the complexity of the model.
> Thank you for this insightful comment. We agree that linear parameter growth is a critical consideration for long-term continual learning. We would like to address this concern from two perspectives: practical storage efficiency and future scalability strategies.
>
> 1. **High Parameter Efficiency:** As detailed in our implementation (Appendix A.1), we utilize LoRA with a rank of $r=32$. Compared to the 7B parameter backbone (LLaVA/InternVL), the storage footprint of a single task-specific LoRA module is extremely marginal (typically **around 1.05% of the base model parameters**). Consequently, even in a scenario with hundreds of tasks, the aggregated storage requirement for these adapters remains significantly smaller than the storage required for a single copy of the base model, making the approach practical for a wide range of real-world deployment scenarios.
> 2. **Mitigation Strategies for Massive Scale:** We acknowledge that for open-ended scenarios involving thousands of tasks, further optimization is necessary. We have identified several promising strategies to mitigate linear growth:
>     - **Expert Merging:** Periodically fusing similar LoRA modules based on the router's similarity scores to reduce the total count of stored experts.
>     - **Hierarchical/Shared Adapters:** Implementing a "library" of shared LoRA bases where new tasks only learn a lightweight delta, rather than a full independent adapter.
>     - **Dynamic Pruning:** Removing or compressing rarely accessed experts over time.
>
> We have updated the **Limitations and Broader Impacts (Appendix D)** section of our paper to explicitly acknowledge the linear growth constraint and discuss these potential mitigation strategies as directions for future work.

---

> > ### Author Response · Authors · 2025-11-26
> >
> > ## Q5: The efficiency of MR-LoRA
> >
> > We appreciate the reviewer raising this important point regarding the potential computational overhead of our two-stage inference process.
> >
> > To address this, we have implemented a specific optimization strategy detailed in Appendix E (Inference Optimization with Caching) to minimize latency. While our method logically involves two phases (routing and prediction), it does not require two full forward passes of the heavy backbone model.
> >
> > Instead, we utilize a caching mechanism where the most computationally intensive operation (the forward pass through the visual encoder and LLM backbone) is performed only once. We cache the resulting hidden states (specifically the KV cache) from this single pass. Subsequently, our lightweight Router and Expert LoRA modules operate sequentially on these cached states.
> >
> > This approach eliminates the need for a second full forward pass, reducing the total computational cost to only marginally more than a single inference. As a result, our deployment cost remains comparable to standard single-pass methods, such as LoRA-FT.

---

### Official Review · Reviewer_x77S · 2025-10-31

**Soundness:** 3
**Presentation:** 3
**Contribution:** 3
**Rating:** 6
**Confidence:** 3

**Summary:**

This paper addresses the critical challenge of catastrophic forgetting and limited plasticity in continual learning scenarios for Multimodal Large Language Models (MLLMs). The authors propose **MR-LoRA**, a novel framework that introduces two key components: (1) training a fresh LoRA adapter from scratch for each new task to preserve model plasticity and avoid negative transfer from previous task weights, and (2) employing a few-shot MLLM-based router that dynamically selects the most suitable expert based on the input modality and query semantics. Additionally, the router is fine-tuned incrementally using a small number of samples from each learned task, enabling it to adapt to new tasks while retaining knowledge of old ones. The method is evaluated on both domain-level and ability-level continual learning benchmarks, demonstrating superior performance in terms of average accuracy, final performance, and backward transfer compared to existing baselines.

**Strengths:**

1.  **Clear and Well-Motivated Problem Formulation:** The paper clearly identifies the dual challenges of stability and plasticity in MLLM continual learning.

2.  **Simple yet Effective Core Idea:** The proposal to train a _fresh_ LoRA from scratch for each task is conceptually simple but effective. This design choice directly tackles the issue of weight interference caused by reusing previous adapters, leading to better new-task performance.

3.  **Innovative Use of MLLM as a Router:** Leveraging the MLLM itself as an intelligent, few-shot router for expert selection is a elegant solution, which is more robust than feature-similarity-based routing.

**Weaknesses:**

1.  **Linear Parameter Growth and Scalability Concerns:** The most significant limitation is the linear increase in the number of stored LoRA modules with the number of tasks. While LoRA is parameter-efficient, storing hundreds or thousands of adapters could become impractical in long-term or open-ended continual learning scenarios. The paper does not discuss potential strategies to mitigate this.

2.  **Limited Discussion on Task Similarity and Negative Transfer:** The paper assumes tasks are distinct enough to warrant separate experts. However, it does not explore scenarios where tasks are highly similar or overlapping. In such cases, having separate LoRAs might lead to redundant learning, and the router might struggle to make consistent decisions.

3.  **Evaluation on Synthetic vs. Real-World Task Sequences:** The experimental setup uses predefined task sequences. A more realistic evaluation would involve open-world scenarios. The generalization capability of the router to completely unseen task categories is not tested.

4. **Novelty Limited:** All content revolves around a central point: adding the LoRA module. Similar approaches exist in continuous learning tasks for standard models. Overall, while effective, the innovation is relatively limited, resembling more of an engineering refinement.

**Questions:**

See Weakness

---

> ### Author Response · Authors · 2025-11-26
>
> ## Q1: Linear Parameter Growth and Scalability Concerns
> Thank you for this insightful comment. We agree that linear parameter growth is a critical consideration for long-term continual learning. We would like to address this concern from two perspectives: practical storage efficiency and future scalability strategies.
>
> 1. **High Parameter Efficiency:** As detailed in our implementation (Appendix A.1), we utilize LoRA with a rank of $r=32$. Compared to the 7B parameter backbone (LLaVA/InternVL), the storage footprint of a single task-specific LoRA module is extremely marginal (typically **around 1.05% of the base model parameters**). Consequently, even in a scenario with hundreds of tasks, the aggregated storage requirement for these adapters remains significantly smaller than the storage required for a single copy of the base model, making the approach practical for a wide range of real-world deployment scenarios.
> 2. **Mitigation Strategies for Massive Scale:** We acknowledge that for open-ended scenarios involving thousands of tasks, further optimization is necessary. We have identified several promising strategies to mitigate linear growth:
>     - **Expert Merging:** Periodically fusing similar LoRA modules based on the router's similarity scores to reduce the total count of stored experts.
>     - **Hierarchical/Shared Adapters:** Implementing a "library" of shared LoRA bases where new tasks only learn a lightweight delta, rather than a full independent adapter.
>     - **Dynamic Pruning:** Removing or compressing rarely accessed experts over time.
>
> We have updated the **Limitations and Broader Impacts (Appendix D)** section of our paper to explicitly acknowledge the linear growth constraint and discuss these potential mitigation strategies as directions for future work.

---

> ### Author Response · Authors · 2025-11-26
>
> ## Q2: Limited Discussion on Task Similarity and Negative Transfer
>
> Thank you for this insightful comment.
>
> Our response is twofold: we will first address the conceptual design of our MR-LoRA framework that handles this scenario, and second, we will present a new empirical study that directly tests this condition.
>
> **Conceptual Design:** The core premise of your question is that the router might "struggle" when faced with two very similar tasks. However, our MLLM-based router is trained not on abstract task labels, but on its ability to select the expert that yields the correct final output.
>
> In the case of two identical or highly similar tasks (e.g., Task A and Task F are the same), we would train two separate experts, expert A and expert F. Because they are trained on the same data distribution, these two experts will become functionally equivalent. When a new input from this task distribution is presented during inference, the router's objective is to pick an expert that maximizes performance. Since both expert A and expert F are equally capable of handling the input, the router can select either one and still achieve a high score.
>
> Therefore, what might seem like a "struggle" or inconsistency in choice is actually a reflection of the task redundancy. The router correctly identifies that multiple experts are suitable, and its final choice does not degrade performance. While this leads to a degree of parameter redundancy (storing two nearly identical experts), it crucially avoids the primary failure mode of other methods: catastrophic forgetting or negative transfer. Our method prioritizes knowledge preservation, and the architectural isolation of experts ensures that learning a redundant task does not corrupt other distinct experts. We will consider the promising strategies discussed in Q1 as our future work to avoid this  parameter redundancy.
>
> **Empirical Validation: Adding a Redundant Task.** To empirically validate this, we have conducted a new experiment as you suggested. We extended the Domain Continual Learning (DCL) setting by adding a sixth task that is an exact repetition of the first task, Remote Sensing (RS).
>
> The new task sequence is: **RS → Medical → Autonomous Driving → Science → Finance → RS (repeat)**.
>
> This creates the exact scenario you described: two tasks are identical, and their corresponding experts should be functionally equivalent. We evaluated the performance after the final task and compared it to the original 5-task DCL setting. The results are shown in the table below.
>
> | **Method** | **RS_1** | **Med** | **AD** | **Sci** | **Fin** | **RS_2** |
> | --- | --- | --- | --- | --- | --- | --- |
> | MR-LoRA (5 tasks) | 80.87 | 65.32 | 54.12 | 56.71 | 91.12 | - |
> | **MR-LoRA (6 tasks)** | 81.02 | 65.83 | 54.17 | 55.94 | 91.11 | 81.02 |
>
> The results clearly demonstrate the robustness of our approach:
>
> 1. **No Catastrophic Forgetting:** The performance on the first Remote Sensing task (**`RS_1`**) remains exceptionally high (81.02 vs. 80.87), showing that learning four intermediate tasks and then relearning RS did not interfere with the original expert.
> 2. **Effective Learning of Redundant Task:** The model achieves high accuracy (81.02) on the newly added sixth task (**`RS_2`**), indicating that it successfully trained a new, effective expert for this repeated domain.
> 3. **Router Stability:** The consistently high performance across all tasks confirms that the router successfully delegates RS inputs to one of the two valid experts without compromising the final prediction. This shows the router does not "struggle" in a way that harms performance.
>
> In summary, our MR-LoRA framework is robust to scenarios with high task similarity. The parameter isolation prevents negative transfer, and the MLLM-based router effectively utilizes the learned experts, even when they are functionally redundant.
>
> We thank you again for this valuable feedback. We have added this experiment and discussion to the revised manuscript (Appendix C.6) to further strengthen our claims.

---

> > ### Author Response · Authors · 2025-11-26
> >
> > ## Q3: Evaluation on Synthetic vs. Real-World Task Sequences
> >
> > We appreciate the reviewer’s insightful suggestion regarding open-world scenarios. We agree that handling completely unseen tasks is a crucial capability for future AI systems. However, we would like to clarify the scope of our work and why the current experimental setup was chosen.
> >
> > **1.  Distinction between Continual Learning and Open-World Learning.** Our paper focuses strictly on **Continual Learning (CL)**, specifically the challenge of sequentially acquiring new domains (DCL) and abilities (ACL) without catastrophic forgetting. The standard protocol for CL research involves evaluating performance on a sequence of tasks that the model is trained on sequentially.
> >
> > - **The Goal of CL:** The primary objective in our setting is to solve the stability-plasticity dilemma, ensuring the model can learn Task B without forgetting Task A.
> > - **The Goal of Open-World Learning:** Evaluating on "completely unseen task categories" typically falls under **Open-World Learning** or **Out-of-Distribution (OOD) Detection**. While highly valuable, this addresses a different problem: detecting novel inputs rather than integrating them into existing knowledge.
> > 1. **Realistic Evaluation within the CL Scope.** While we do not cover open-world scenarios, we respectfully argue that **MLLM-CL** is already significantly more realistic than existing CL benchmarks.
> > - Prior works often rely on simple class-incremental learning within a single dataset (IID).
> > - In contrast, our **Ability Continual Learning (ACL)** setting specifically evaluates on **non-IID** scenarios (training on one dataset, evaluating on a different one with shifted distributions). This tests the generalization of the *learned* abilities effectively, which is a major step forward in making CL evaluation more practical.
> >
> > In summary, while open-world evaluation is an exciting direction for future research, it is outside the scope of this paper, which is dedicated to establishing a robust benchmark and method for Continual Learning in MLLMs. We have added a discussion in the “Future Work” (Appendix D.2) section to acknowledge open-world scenarios as a promising avenue for future work.

---

> > > ### Author Response · Authors · 2025-11-26
> > >
> > > ## Q4: Novelty Limited
> > >
> > > We thank the reviewer for their feedback and for acknowledging the effectiveness of our approach. However, we respectfully disagree with the assessment that the novelty is limited to "adding a LoRA module." We believe there may be a misunderstanding regarding the scope of our contributions.
> > >
> > > Our work provides two distinct and significant contributions to the field: **1) A novel, comprehensive benchmark (MLLM-CL)**, and **2) A novel routing mechanism (MR-LoRA)** that fundamentally differs from standard parameter isolation methods
> > >
> > > **1. Novelty of the Benchmark (MLLM-CL).** A primary contribution of this paper is the establishment of **MLLM-CL**, which addresses critical gaps in existing research.
> > >
> > > - **Beyond Dataset Incremental Learning:** Prior benchmarks largely focus on simple dataset incremental learning where training and test sets are IID (Identically and Independently Distributed).
> > > - **Two Practical Settings:** We introduce two distinct settings: **Domain Continual Learning (DCL)** for accumulating domain-specific knowledge (Medical, Finance, Remote Sensing), and **Ability Continual Learning (ACL)** for acquiring fundamental skills (OCR, Logic, GUI Agents) evaluated on **non-IID** scenarios.
> > > - **New Data Pipelines:** We constructed specific datasets to enable this, such as the *StockQA* dataset generated via a novel questioner-inspector pipeline (Section 3), to ensure rigorous evaluation across diverse real-world domains. This benchmark provides the community with a necessary standard for evaluating lifelong learning in MLLMs, which did not exist previously.
> > >
> > > **2. Novelty of the Method (MR-LoRA).** While we utilize LoRA as the efficient tuning module, our methodological innovation lies in the **MLLM-based Routing Mechanism**, not the LoRA adaptation itself.
> > >
> > > - **Semantic vs. Heuristic Routing:** Existing parameter isolation methods rely on hand-crafted, shallow metrics like cosine similarity of embeddings to select experts. These heuristics often fail with complex multimodal data.
> > > - **Intrinsic Routing:** Our approach (MR-LoRA) transforms the routing problem by leveraging the MLLM’s *intrinsic* multimodal understanding capabilities. We tune a lightweight router to semantically analyze the input and select the optimal expert (Figure 5). This is not a standard CL application; it is a new paradigm for expert selection that leverages the "intelligence" of the model itself rather than external metrics.
> > > - **Superior Performance:** This is not merely an engineering tweak; the impact is substantial. As shown in Tables 2 and 3, MR-LoRA achieves near-Oracle performance and significantly outperforms standard LoRA-based CL methods (like O-LoRA and MoELoRA), proving that *how* the modules are selected is just as critical as the modules themselves.
> > >
> > > In summary, we go beyond simple parameter efficient tuning. We propose a foundational benchmark for the field and a routing methodology that solves the catastrophic interference problem more effectively than existing heuristic approaches.

---

### Author Response · Authors · 2025-12-03
**Summary of Rebuttal**

Dear AC,

We sincerely thank all reviewers for their constructive feedback and appreciate your time and effort in handling our paper. Below, we summarize the main points from the discussion.

---
**Acknowledged strengths:**
- Clear, well-motivated problem: reviewers agreed the paper addresses an important, timely challenge (continual learning for Multimodal LLMs) and frames the stability–plasticity trade-off clearly.
- A useful benchmark: the proposed MLLM-CL (Domain CL + Ability CL) is seen as a valuable step toward standardized evaluation for continual learning in multimodal LLMs.
- Innovative routing design: using the MLLM itself as a few‑shot semantic router (rather than simple feature-similarity heuristics) was highlighted as an elegant and effective contribution.
- Strong empirical results: MR-LoRA achieves near-oracle performance on DCL and significant gains on ACL; multiple reviewers noted solid experimental performance relative to baselines.
- Good presentation and writing: several reviewers found the paper well-written, easy to follow, with useful figures and ablation studies.

---
**Following the advice of reviewers, we have added these discussions and experiments:**
- Discussion about training-free methods, justification  of DCL and ACL, the inference efficiency of MR-LoRA, the scalability, complexities and number of trained parameters, and the order-agnostic nature of our method.
- Experiments on more benchmarks (UCIT) and more comparison methods (L2P, ModalPrompt), and empirical validation on adding two similar tasks in CL process.

---
**We discuss several points of interest raised by the reviewers that are either beyond the scope of the present study or based on potential misunderstandings.**
- Distinction between Continual Learning and Open-World Learning. Our paper focuses strictly on Continual Learning (CL), specifically the challenge of sequentially acquiring new domains (DCL) and abilities (ACL) without catastrophic forgetting. The standard protocol for CL research involves evaluating performance on a sequence of tasks that the model is trained on sequentially. However, evaluating on "completely unseen task categories" typically falls under Open-World Learning or Out-of-Distribution (OOD) Detection.
- Potentially overlooked contribution of this paper. Our work provides two distinct and significant contributions to the field: 1) A novel, comprehensive benchmark (MLLM-CL), and 2) A novel routing mechanism (MR-LoRA) that fundamentally differs from standard parameter isolation methods. Some reviewers miss the contribution of our benchmark.

Finally, there are some minor suggestions from reviewers that we believe should not be grounds for rejection, like the open source of code in the paper submission stage and adding standard deviation values.

---
In summary, we have diligently addressed every point raised by the reviewers, strengthening the paper with new experiments, discussions, and clarifications. **Reviewer AV79 is willing to increase the rating in the review.** We have also cataloged valuable ideas that fall outside this paper’s scope as priorities for our future research.
We are deeply grateful for the time and expertise invested by the reviewers and the Area Chair. We believe the manuscript is now significantly improved and hope it earns your support.

---

### Meta-Review · Area_Chair_4vFj · 2025-12-15

**Summary:**

The paper aims at addressing the challenges of catastrophic forgetting and limited plasticity when continuously adapting Multimodal Large Language Models for new tasks. The paper presented a new MLLM-CL benchmark and a new method based on a LoRA adapter per new task and a few-shot MLLM-based router.

Strengths identified by reviewers include, its writing, well-motivation, simple method, and contributed dataset.

However, the paper also faces several critical weaknesses that have not been fully addressed in the rebuttal. The novelty of the proposed method has been questioned, with reviewers pointing out that the idea of learning task-specific LoRA adapter (expert) and router to select experts has been explored in continuous learning tasks for standard models. Additionally, continuously adapting to new domains or new abilities would increase the complexity of the model and the proposed two-stage inference would also increase the computational cost.

Given these considerations, this paper fails to meet the standards of ICLR. The authors are encouraged to address the highlighted issues to strengthen their contribution to the field of Multimodal Large Language Model Continual Learning.

**Reviewer Concerns:**

The authors provided detailed feedback with additional experiments and revisions in the manuscript. Most of concerns have been addressed, including, 1) limited discussion on task similarity and negative transfer, comparisons to training-free methods like ICL, novelty of the introduced benchmark MLLM-CL, comparisons with more recent methods, effectiveness of the router when tasks do not have clear domain boundaries and stable task definitions, results on other benchmarks, learning task-agnostic experts or router, effectiveness of method with permutation of the tasks, more metrics, and issues with presentation.

However, several concerns are not fully addressed including, limited novelty of the method (Lora and expert selection are commonly used strategies), increasing complexity, parameters, computational cost when continuously adapting to new tasks, effectiveness on unseen tasks, and limited generality for practical applications (unseen domains or real-time streaming data).

**Reviewer Scores:**

The initial reviewer scores are mixed (three borderline reject, and one borderline accept). Reviewer AV79 would have raised the score while other reviewers would have maintained their scores.

---

### Decision · Program_Chairs · 2026-01-26

Reject